# Massively Scalable Sinkhorn Distances
# via the Nyström Method

**Jason Altschuler**
MIT
jasonalt@mit.edu

**Francis Bach**
INRIA - ENS - PSL
francis.bach@inria.fr

**Alessandro Rudi**
INRIA - ENS - PSL
alessandro.rudi@inria.fr

**Jonathan Niles-Weed**
NYU
jnw@cims.nyu.edu

## Abstract

The Sinkhorn "distance," a variant of the Wasserstein distance with entropic regularization, is an increasingly popular tool in machine learning and statistical inference. However, the time and memory requirements of standard algorithms for computing this distance grow quadratically with the size of the data, making them prohibitively expensive on massive data sets. In this work, we show that this challenge is surprisingly easy to circumvent: combining two simple techniques—the Nyström method and Sinkhorn scaling—provably yields an accurate approximation of the Sinkhorn distance with significantly lower time and memory requirements than other approaches. We prove our results via new, explicit analyses of the Nyström method and of the stability properties of Sinkhorn scaling. We validate our claims experimentally by showing that our approach easily computes Sinkhorn distances on data sets hundreds of times larger than can be handled by other techniques.

## 1 Introduction

Optimal transport is a fundamental notion in probability theory and geometry [42], which has recently attracted a great deal of interest in the machine learning community as a tool for image recognition [26, 35], domain adaptation [11, 12], and generative modeling [5, 9, 20], among many other applications [see, e.g., 25, 31].

The growth of this field has been fueled in part by computational advances, many of them stemming from an influential proposal of Cuturi [13] to modify the definition of optimal transport to include an *entropic penalty*. The resulting quantity, which Cuturi [13] called the *Sinkhorn "distance"*[1] after Sinkhorn [38], is significantly faster to compute than its unregularized counterpart. Though originally attractive purely for computational reasons, the Sinkhorn distance has since become an object of study in its own right because it appears to possess better statistical properties than the unregularized distance both in theory and in practice [21, 29, 31, 34, 36]. Computing this distance as quickly as possible has therefore become an area of active study.

We briefly recall the setting. Let $\mathbf{p}$ and $\mathbf{q}$ be probability distributions supported on at most $n$ points in $\mathbb{R}^d$. We denote by $\mathcal{M}(\mathbf{p}, \mathbf{q})$ the set of all *couplings* between $\mathbf{p}$ and $\mathbf{q}$, and for any $P \in \mathcal{M}(\mathbf{p}, \mathbf{q})$, we denote by $H(P)$ its Shannon entropy. (See Section 2.1 for full definitions.) The Sinkhorn distance

between **p** and **q** is defined as

$$W_\eta(\mathbf{p}, \mathbf{q}) := \min_{P \in \mathcal{M}(\mathbf{p}, \mathbf{q})} \sum_{ij} P_{ij} \|x_i - x_j\|_2^2 - \eta^{-1} H(P), \tag{1}$$

for a parameter $\eta > 0$. We stress that we use the squared Euclidean cost in our formulation of the Sinkhorn distance. This choice of cost—which in the unregularized case corresponds to what is called the 2-Wasserstein distance [42]—is essential to our results, and we do not consider other costs here. The squared Euclidean cost is among the most common in applications [9, 12, 16, 21, 36].

Many algorithms to compute $W_\eta(\mathbf{p}, \mathbf{q})$ are known. Cuturi [13] showed that a simple iterative procedure known as Sinkhorn's algorithm had very fast performance in practice, and later experimental work has shown that greedy and stochastic versions of Sinkhorn's algorithm perform even better in certain settings [3, 20]. These algorithms are notable for their versatility: they provably succeed for any bounded, nonnegative cost. On the other hand, these algorithms are based on matrix manipulations involving the $n \times n$ cost matrix $C$, so their running times and memory requirements inevitably scale with $n^2$. In experiments, Cuturi [13] and Genevay et al. [20] showed that these algorithms could reliably be run on problems of size $n \approx 10^4$.

Another line of work has focused on obtaining better running times when the cost matrix has special structure. A preeminent example is due to Solomon et al. [40], who focus on the Wasserstein distance on a compact Riemannian manifold, and show that an approximation to the entropic regularized Wasserstein distance can be obtained by repeated convolution with the heat kernel on the domain. Solomon et al. [40] also establish that for data supported on a grid in $\mathbb{R}^d$, significant speedups are possible by decomposing the cost matrix into "slices" along each dimension [see 31, Remark 4.17]. While this approach allowed Sinkhorn distances to be computed on significantly larger problems ($n \approx 10^8$), it does not extend to non-grid settings. Other proposals include using random sampling of auxiliary points to approximate semi-discrete costs [41] or performing a Taylor expansion of the kernel matrix in the case of the squared Euclidean cost [4]. These approximations both focus on the $\eta \to \infty$ regime, when the regularization term in (1) is very small, and do not apply to the moderately regularized case $\eta = O(1)$ typically used in practice. Moreover, the running time of these algorithms scales exponentially in the ambient dimension, which can be very large in applications.

## 1.1 Our contributions

We show that a simple algorithm can be used to approximate $W_\eta(\mathbf{p}, \mathbf{q})$ quickly on massive data sets. Our algorithm uses only known tools, but we give novel theoretical guarantees that allow us to show that the Nyström method combined with Sinkhorn scaling *provably* yields a valid approximation algorithm for the Sinkhorn distance at a fraction of the running time of other approaches.

We establish two theoretical results of independent interest: **(i)** New Nyström approximation results showing that instance-adaptive low-rank approximations to Gaussian kernel matrices can be found for data lying on a low-dimensional manifold (Section 3.) **(ii)** New stability results about Sinkhorn projections, establishing that a sufficiently good approximation to the cost matrix can be used (Section 4.)

## 1.2 Prior work

Computing the Sinkhorn distance efficiently is a well studied problem in a number of communities. The Sinkhorn distance is so named because, as was pointed out by Cuturi [13], there is an extremely simple iterative algorithm due to Sinkhorn [38] which converges quickly to a solution to (1). This algorithm, which we call *Sinkhorn scaling*, works very well in practice and can be implemented using only matrix-vector products, which makes it easily parallelizable. Sinkhorn scaling has been analyzed many times [3, 14, 17, 24, 27], and forms the basis for the first algorithms for the unregularized optimal transport problem that run in time nearly linear in the size of the cost matrix [3, 14]. Greedy and stochastic algorithms related to Sinkhorn scaling with better empirical performance have also been explored [3, 20]. Another influential technique, due to Solomon et al. [40], exploits the fact that, when the distributions are supported on a grid, Sinkhorn scaling performs extremely quickly by decomposing the cost matrix along lower-dimensional slices.

Other algorithms have sought to solve (1) by bypassing Sinkhorn scaling entirely. Blanchet et al. [8] proposed to solve (1) directly using second-order methods based on fast Laplacian solvers [2, 10].

Blanchet et al. [8] and Quanrud [32] have noted a connection to packing linear programs, which can also be exploited to yield near-linear time algorithms for unregularized transport distances.

Our main algorithm relies on constructing a low-rank approximation of a Gaussian kernel matrix from a small subset of its columns and rows. Computing such approximations is a problem with an extensive literature in machine learning, where it has been studied under many different names, e.g., Nyström method [44], sparse greedy approximations [39], incomplete Cholesky decomposition [15], Gram-Schmidt orthonormalization [37] or CUR matrix decompositions [28]. The approximation properties of these algorithms are now well understood [1, 6, 22, 28]; however, in this work, we require significantly more accurate bounds than are available from existing results as well as adaptive bounds for low-dimensional data. To establish these guarantees, we follow an approach based on approximation theory [see, e.g., 7, 33, 43], which consists of analyzing interpolation operators for the reproducing kernel Hilbert space corresponding to the Gaussian kernel.

Finally, this paper adds to recent work proposing the use of low-rank approximation for Sinkhorn scaling [4, 41]. We improve upon those papers in several ways. First, although we also exploit the idea of a low-rank approximation to the kernel matrix, we do so in a more sophisticated way that allows for automatic adaptivity to data with low-dimensional structure. These new approximation results are the key to our adaptive algorithm, and this yields a significant improvement in practice. Second, the analyses of Altschuler et al. [4] and Tenetov et al. [41] only yield an approximation to $W_\eta(\mathbf{p}, \mathbf{q})$ when $\eta \to \infty$. In the moderately regularized case when $\eta = O(1)$, which is typically used in practice, neither the work of Altschuler et al. [4] nor of Tenetov et al. [41] yields a rigorous error guarantee.

## 2 Main Result

### 2.1 Preliminaries and notation

**Problem setup.** Throughout, $\mathbf{p}$ and $\mathbf{q}$ are two probability distributions supported on a set $X := \{x_1, \ldots, x_n\}$ of points in $\mathbb{R}^d$, with $\|x_i\|_2 \leqslant R$ for all $i \in [n] := \{1, \ldots, n\}$. We define the cost matrix $C \in \mathbb{R}^{n \times n}$ by $C_{ij} = \|x_i - x_j\|_2^2$. We identify $\mathbf{p}$ and $\mathbf{q}$ with vectors in the simplex $\Delta_n := \{v \in \mathbb{R}_{\geqslant 0}^n : \sum_{i=1}^n v_i = 1\}$ whose entries denote the weight each distribution gives to the points of $X$. We denote by $\mathcal{M}(\mathbf{p}, \mathbf{q})$ the set of couplings between $\mathbf{p}$ and $\mathbf{q}$, identified with the set of $P \in \mathbb{R}_{\geqslant 0}^{n \times n}$ satisfying $P\mathbf{1} = \mathbf{p}$ and $P^\top \mathbf{1} = \mathbf{q}$, where $\mathbf{1}$ denotes the all-ones vector in $\mathbb{R}^n$. The Shannon entropy of a non-negative matrix $P \in \mathbb{R}_{\geqslant 0}^{n \times n}$ is denoted $H(P) := \sum_{ij} P_{ij} \log \frac{1}{P_{ij}}$, where we adopt the standard convention that $0 \log \frac{1}{0} = 0$.

Our goal is to approximate the Sinkhorn distance (1) to some additive accuracy $\varepsilon > 0$. By strict convexity, this optimization problem has a unique minimizer, which we denote henceforth by $P^\eta$. For shorthand, in the sequel we write

$$V_M(P) := \langle M, P \rangle - \eta^{-1} H(P),$$

for a matrix $M \in \mathbb{R}^{n \times n}$. In particular, we have $W_\eta(\mathbf{p}, \mathbf{q}) = \min_{P \in \mathcal{M}(\mathbf{p}, \mathbf{q})} V_C(P)$. For the purpose of simplifying some bounds, we assume throughout that $n \geqslant 2$, $\eta \in [1, n]$, $R \geqslant 1$, $\varepsilon \leqslant 1$.

**Sinkhorn scaling.** Our approach is based on Sinkhorn scaling, an algorithm due to Sinkhorn [38] and popularized for optimal transport by Cuturi [13]. We recall the following fundamental definition.

**Definition 1.** Given $\mathbf{p}, \mathbf{q} \in \Delta_n$ and $K \in \mathbb{R}^{n \times n}$ with positive entries, the *Sinkhorn projection* $\Pi_{\mathcal{M}(\mathbf{p}, \mathbf{q})}^{\mathcal{S}}(K)$ of $K$ onto $\mathcal{M}(\mathbf{p}, \mathbf{q})$ is the *unique* matrix in $\mathcal{M}(\mathbf{p}, \mathbf{q})$ of the form $D_1 K D_2$ for positive diagonal matrices $D_1$ and $D_2$.

Since $\mathbf{p}$ and $\mathbf{q}$ remain fixed throughout, we abbreviate $\Pi_{\mathcal{M}(\mathbf{p}, \mathbf{q})}^{\mathcal{S}}$ by $\Pi^{\mathcal{S}}$ except when we want to make the feasible set $\mathcal{M}(\mathbf{p}, \mathbf{q})$ explicit.

**Proposition 1** (45)**.** *Let $K$ have strictly positive entries, and let $\log K$ be the matrix defined by* $(\log K)_{ij} := \log(K_{ij})$. *Then*

$$\Pi_{\mathcal{M}(\mathbf{p}, \mathbf{q})}^{\mathcal{S}}(K) = \operatorname*{argmin}_{P \in \mathcal{M}(\mathbf{p}, \mathbf{q})} \langle -\log K, P \rangle - H(P).$$

*Note that the strict convexity of $-H(P)$ and the compactness of $\mathcal{M}(\mathbf{p}, \mathbf{q})$ implies that the minimizer exists and is unique.*

This yields the following simple but key connection between Sinkhorn distances and Sinkhorn scaling.

**Corollary 1.**

$$P^\eta = \Pi^{\mathcal{S}}_{\mathcal{M}(\mathbf{p}, \mathbf{q})}(K) \,,$$

*where $K$ is defined by $K_{ij} = e^{-\eta C_{ij}}$.*

**Notation.** We define the probability simplices $\Delta_n := \{p \in \mathbb{R}^n_{\geqslant 0} : p^\top \mathbf{1} = 1\}$ and $\Delta_{n \times n} := \{P \in \mathbb{R}^{n \times n}_{\geqslant 0} : \mathbf{1}^\top P \mathbf{1} = 1\}$. Elements of $\Delta_{n \times n}$ will be called *joint distributions*. The Kullback-Leibler divergence between two joint distributions $P$ and $Q$ is $\mathsf{KL}(P \| Q) := \sum_{ij} P_{ij} \log \frac{P_{ij}}{Q_{ij}}$.

Throughout the paper, all matrix exponentials and logarithms will be taken entrywise, i.e., $(e^A)_{ij} := e^{A_{ij}}$ and $(\log A)_{ij} := \log A_{ij}$ for $A \in \mathbb{R}^{n \times n}$. Given a matrix $A$, we denote by $\|A\|_{\mathrm{op}}$ its operator norm (i.e., largest singular value), by $\|A\|_*$ its nuclear norm (i.e., the sum of its singular values), by $\|A\|_1$ its entrywise $\ell_1$ norm (i.e., $\|A\|_1 := \sum_{ij} |A_{ij}|$), and by $\|A\|_\infty$ its entrywise $\ell_\infty$ norm (i.e., $\|A\|_\infty := \max_{ij} |A_{ij}|$). We abbreviate "positive semidefinite" by "PSD."

The notation $f = O(g)$ means that $f \leqslant Cg$ for some universal constant $C$, and $g = \Omega(f)$ means $f = O(g)$. The notation $\tilde{O}(\cdot)$ omits polylogarithmic factors depending on $R$, $\eta$, $n$, and $\varepsilon$.

## 2.2 Main result and proposed algorithm

Pseudocode for our proposed algorithm is given in Algorithm 1. NYS-SINK (pronounced "nice sink") computes a low-rank Nyström approximation of the kernel matrix via a column sampling procedure. For reasons of space, full pseudocode and proofs of all claims are deferred to the supplement.

As noted in Section 1, the Nyström method constructs a low-rank approximation to a Gaussian kernel matrix $K = e^{-\eta C}$ based on a small number of its columns. In order to design an efficient algorithm, we aim to construct such an approximation with the smallest possible rank. The key quantity for understanding the error of this algorithm is the so-called *effective dimension* (also sometimes called the "degrees of freedom") of the kernel matrix $K$ [18, 30, 46].

**Definition 2.** Let $\lambda_j(K)$ denote the $j$th largest eigenvalue of $K$ (with multiplicity). Then the *effective dimension* of $K$ at level $\tau > 0$ is

$$d_{\mathrm{eff}}(\tau) := \sum_{j=1}^n \frac{\lambda_j(K)}{\lambda_j(K) + \tau n}. \tag{2}$$

The effective dimension $d_{\mathrm{eff}}(\tau)$ indicates how large the rank of an approximation $\tilde{K}$ to $K$ must be in order to obtain the guarantee $\|\tilde{K} - K\|_{\mathrm{op}} \leqslant \tau n$. For our application, we have $K = e^{\eta C}$, and we will show that it suffices to obtain an approximate kernel $\tilde{K}$ satisfying $\|\tilde{K} - K\|_{\mathrm{op}} \leqslant \frac{\varepsilon'}{2} e^{-4\eta R^2}$, where $\varepsilon' = \tilde{O}(\varepsilon R^{-2})$. We are therefore motivated to define the following quantity, which informally captures the smallest possible rank of an approximation of this quality.

**Definition 3.** Given $X = \{x_1, \ldots, x_n\} \subseteq \mathbb{R}^d$ with $\|x_i\|_2 \leqslant R$ for all $i \in [n]$, $\eta > 0$, and $\varepsilon' \in (0, 1)$, the *approximation rank* is

$$r^*(X, \eta, \varepsilon') := d_{\mathrm{eff}}\left(\frac{\varepsilon'}{2n} e^{-4\eta R^2}\right)$$

where $d_{\mathrm{eff}}(\cdot)$ is the effective rank for the kernel matrix $K := e^{-\eta C}$.

As we show below, we adaptively construct an approximate kernel $\tilde{K}$ whose rank is at most a logarithmic factor bigger than $r^*(X, \eta, \varepsilon')$ with high probability. We also give concrete bounds on $r^*(X, \eta, \varepsilon')$ below.

Our proposed algorithm makes use of several subroutines. The ADAPTIVENYSTRÖM procedure in Algorithm 1 combines an algorithm of Musco & Musco [30] with a doubling trick that enables automatic adaptivity. It outputs the approximate kernel $\tilde{K}$ and its rank $r$.

The SINKHORN procedure in Algorithm 1 is the Sinkhorn scaling algorithm for projecting $\tilde{K}$ onto $\mathcal{M}(\mathbf{p}, \mathbf{q})$. We use a variant of the standard algorithm, which returns both the scaling matrices and an approximation of the cost of an optimal solution. The ROUND procedure in Algorithm 1 is Algorithm 2 of Altschuler et al. [3].

We emphasize that neither $D_1 \tilde{K} D_2$ nor $\hat{P}$ are ever represented explicitly, since this would take $\Omega(n^2)$ time. Instead, we maintain these matrices in low-rank factorized forms. This enables Algorithm 1 to be implemented efficiently in $o(n^2)$ time, since the procedures SINKHORN and ROUND can both be implemented such that they depend on $\tilde{K}$ only through matrix-vector multiplications with $\tilde{K}$. Moreover, we also emphasize that all steps of Algorithm 1 are easily parallelizable since they can be re-written in terms of matrix-vector multiplications.

We note also that although the present paper focuses specifically on the squared Euclidean cost $c(x_i, x_j) = \|x_i - x_j\|_2^2$ (corresponding to the 2-Wasserstein case of optimal transport pervasively used in applications; see intro), our algorithm NYS-SINK readily extends to other cases of optimal transport. Indeed, since the Nyström method works not only for Gaussian kernel matrices $K_{ij} = e^{-\eta \|x_i - x_j\|_2^2}$, but in fact more generally for any PSD kernel matrix, our algorithm can be used on any optimal transport instance for which the corresponding kernel matrix $K_{ij} = e^{-\eta c(x_i, x_j)}$ is PSD.

---

**Algorithm 1** NYS-SINK

---

1: **Input:** $X = \{x_1, \ldots, x_n\} \subseteq \mathbb{R}^d$, $\mathbf{p}, \mathbf{q} \in \Delta_n$, $\varepsilon, \eta > 0$
2: **Output:** $\hat{P} \in \mathcal{M}(\mathbf{p}, \mathbf{q})$, $\hat{W} \in \mathbb{R}$, $r \in \mathbb{N}$
3: $\varepsilon' \leftarrow \min(1, \frac{\varepsilon\eta}{50(4R^2\eta + \log \frac{n}{\eta\varepsilon})})$
4: $(\tilde{K}, r) \leftarrow$ ADAPTIVENYSTRÖM$(X, \eta, \frac{\varepsilon'}{2} e^{-4\eta R^2})$
   {Compute low-rank approximation}
5: $(D_1, D_2, \hat{W}) \leftarrow$ SINKHORN$(\tilde{K}, \mathbf{p}, \mathbf{q}, \varepsilon')$
   {Approximate Sinkhorn projection and cost}
6: $\hat{P} \leftarrow$ ROUND$(D_1 \tilde{K} D_2, \mathbf{p}, \mathbf{q})$
   {Round to feasible set}
7: Return $\hat{P}, \hat{W}$

---

Our main result is the following.

**Theorem 1.** *Let $\varepsilon, \delta \in (0, 1)$. Algorithm 1 runs in $\tilde{O}\left(nr\left(r + \frac{\eta R^4}{\varepsilon}\right)\right)$ time, uses $O(n(r+d))$ space, and returns a feasible matrix $\hat{P} \in \mathcal{M}(\mathbf{p}, \mathbf{q})$ in factored form and scalars $\hat{W} \in \mathbb{R}$ and $r \in \mathbb{N}$, where*

$$|V_C(\hat{P}) - W_\eta(\mathbf{p}, \mathbf{q})| \leqslant \varepsilon, \tag{3a}$$

$$\mathsf{KL}(\hat{P}\|P^\eta) \leqslant \eta\varepsilon, \tag{3b}$$

$$|\hat{W} - W_\eta(\mathbf{p}, \mathbf{q})| \leqslant \varepsilon, \tag{3c}$$

*and, with probability $1 - \delta$,*

$$r \leqslant c \cdot r^*(X, \eta, \varepsilon') \log \frac{n}{\delta}, \tag{3d}$$

*for a universal constant $c$ and where $\varepsilon' = \tilde{\Omega}(\varepsilon R^{-2})$.*

We note that, while our algorithm is randomized, we obtain a deterministic guarantee that $\hat{P}$ is a good solution. We also note that runtime dependence on the radius $R$—which governs the scale of the problem—is inevitable since we seek an additive guarantee.

We show in Section 3 that $r^*$—which controls the running time of the algorithm with high probability by (3d)—adapts to the *intrinsic* dimension of the data. This adaptivity is crucial in applications, where data can have much lower dimension than the ambient space. We informally summarize this behavior in the following theorem.

**Theorem 2** (Informal)**.** *There exists an universal constant $c > 0$ such that, for any $n$ points in a ball of radius $R$ in $\mathbb{R}^d$, $r^*(X, \eta, \varepsilon') \leqslant (c(\eta R^2 + \log \frac{n}{\varepsilon'\eta}))^d$. Moreover, for any $k$-dimensional manifold $\Omega$ satisfying certain technical conditions and $\eta > 0$, there exists a constant $c_{\Omega, \eta}$ such that for any $n$ points lying on $\Omega$, $r^*(X, \eta, \varepsilon') \leqslant c_{\Omega, \eta}(\log \frac{n}{\varepsilon'})^{5k/2}$.*

The formal versions of these bounds appear in Section 3. The second bound is significantly better than the first when $k \ll d$, and clearly shows the benefits of an adaptive procedure.

Combining Theorems 1 and 2 yields the following time and space complexity for our algorithm.

**Corollary 2** (Informal). *If $X$ consists of $n$ points lying in a ball of radius $R$ in $\mathbb{R}^d$, then with high probability Algorithm 1 requires $\tilde{O}(n\varepsilon^{-1}(c\eta R^2 + c\log \frac{n}{\varepsilon})^{2d+1})$ time and $\tilde{O}(n(c\eta R^2 + c\log \frac{n}{\varepsilon})^d)$ space. Moreover, if $X$ lies on a $k$-dimensional manifold $\Omega$, then with high probability Algorithm 1 requires $\tilde{O}(n\varepsilon^{-1}c_{\Omega,\eta}(\log \frac{n}{\varepsilon})^{5k})$ time and $\tilde{O}(nc_{\Omega,\eta}(\log \frac{n}{\varepsilon})^{5k/2})$ space.*

## 3 Kernel Approximation via the Nyström Method

Given points $X = \{x_1, \ldots, x_n\}$ with $\|x_i\|_2 \leqslant R$ for all $i \in [n]$, let $K \in \mathbb{R}^{n \times n}$ denote the matrix with entries $K_{ij} := k_\eta(x_i, x_j)$, where $k_\eta(x, x') := e^{-\eta\|x-x'\|^2}$. Note that $k_\eta(x, x')$ is the Gaussian kernel $e^{-\|x-x'\|^2/(2\sigma^2)}$ between points $x$ and $x'$ with bandwith parameter $\sigma^2 = \frac{1}{2\eta}$. For $r \in \mathbb{N}$, we consider an approximation of the matrix $K$ that is of the form $\widetilde{K} = VA^{-1}V^\top$, where $V \in \mathbb{R}^{n \times r}$ and $A \in \mathbb{R}^{r \times r}$. Note that the matrix $\widetilde{K}$ is never computed explicitly. Indeed, our proposed Algorithm 1 only depends on $\widetilde{K}$ through computing matrix-vector products $\widetilde{K}v$, where $v \in \mathbb{R}^n$, and these can be computed efficiently as $\widetilde{K}v = V(L^{-\top}(L^{-1}(V^\top v)))$, where $L \in \mathbb{R}^{r \times r}$ is the lower triangular matrix satisfying $LL^\top = A$. Once a Cholesky decomposition of $A$ has been obtained—at computational cost $O(r^3)$—matrix-vector products can therefore be computed in time $O(nr)$. In the supplement, we give pseudocode for the AdaptiveNyström subroutine, based on a simple doubling trick. It enjoys the following guarantee:

**Lemma 1.** *Let $\tilde{K}$ denote the (random) kernel output by $\textsc{AdaptiveNyström}(X, \eta, \tau)$, and let $r := rank(\tilde{K})$. Then $\|K - \tilde{K}\|_\infty \leqslant \tau$, the algorithm used $O(nr)$ space and terminated in $O(nr^2)$ time, and there exists a universal constant $c$ such that simultaneously for every $\delta > 0$,*

$$\mathbb{P}\left(r \leqslant c \cdot d_{\mathit{eff}}\left(\tfrac{\tau}{n}\right)\log\left(\tfrac{n}{\delta}\right)\right) \geqslant 1 - \delta.$$

### 3.1 General results: data points lie in a ball

In this section we assume no structure on $X$ apart from the fact that $X \subseteq B_R^d$ where $B_R^d$ is a ball of radius $R$ in $\mathbb{R}^d$ centered around the origin, for some $R > 0$ and $d \in \mathbb{N}$. First we characterize the eigenvalues of $K$ in terms of $\eta, d, R$, and then we use this to bound $d_{\text{eff}}$.

**Theorem 3.** *Let $X := \{x_1, \ldots x_n\} \subseteq B_R^d$, and let $K \in \mathbb{R}^{n \times n}$ be the matrix with entries $K_{ij} := e^{-\eta\|x_i - x_j\|^2}$. Then: **1.** For $t \geqslant (2e)^d$, $\lambda_{t+1}(K) \leqslant ne^{-\frac{d}{2e}t^{1/d}\log\frac{d\,t^{1/d}}{4e^2\eta R^2}}$. **2.** For $\tau \in (0,1]$, $d_{\mathit{eff}}(\tau) \leqslant 3\left(6 + \frac{41}{d}\eta R^2 + \frac{3}{d}\log\frac{1}{\tau}\right)^d$.*

**Corollary 3.** *Let $\varepsilon' \in (0,1)$ and $\eta > 0$. If $X$ consists of $n$ points lying in a ball of radius $R$ around the origin in $\mathbb{R}^d$, then*

$$r^*(X, \eta, \varepsilon') \leqslant 3\left(6 + \frac{53}{d}\eta R^2 + \frac{3}{d}\log\frac{2n}{\varepsilon'}\right)^d.$$

### 3.2 Adaptivity: data points lie on a low dimensional manifold

In this section, we show that the quality of the Nyström approximation adapts to the intrinsic dimension of the data. Let $\Omega \subset \mathbb{R}^d$ be a smooth compact manifold without boundary of dimension $k$, for $k < d$, and let $(\Psi_j, U_j)_{j \in [T]}$, with $T \in \mathbb{N}$, be an atlas for $\Omega$. We assume the following quantitative control on the smoothness of the atlas.

**Assumption 1.** There exists $Q > 0$ such that

$$\sup_{u \in B_{r_j}^k} \|D^\alpha \Psi_j^{-1}(u)\| \leqslant Q^{|\alpha|}, \qquad \alpha \in \mathbb{N}^k, j \in [T],$$

where $|\alpha| = \sum_{j=1}^k \alpha_j$ and $D^\alpha = \frac{\partial^{|\alpha|}}{\partial u_1^{\alpha_1}\ldots\partial u_k^{\alpha_k}}$, for $\alpha \in \mathbb{N}^k$.

**Theorem 4.** *Let $\Omega \subset B_R^d \subset \mathbb{R}^d$ be a smooth compact manifold without boundary satisfying Assumption 1. Let $X := \{x_1, \ldots x_n\} \subseteq \Omega$, and let $K \in \mathbb{R}^{n \times n}$ be the matrix with entries $K_{ij} := e^{-\eta \|x_i - x_j\|^2}$. Then: **1.** There exists a constant $c$ not depending on $X$ or $n$, such that for $t \geqslant 0$, $\lambda_{t+1}(K) \leqslant n e^{-ct^{\frac{2}{5k}}}$. **2.** There exist $c_1, c_2$ not depending on $X$, $n$, or $\tau$, such that for $\tau \in (0, 1]$, $d_{eff}(\tau) \leqslant \left( c_1 \log \frac{1}{\tau} \right)^{5k/2} + c_2$.*

The result above is new, to our knowledge, and extends interpolation results on manifolds [19, 23, 43], from polynomial to exponential decay, generalizing a technique of Rieger & Zwicknagl [33] to a subset of real analytic manifolds. The crucial point is that now the eigenvalue decay and the effective dimension depend on the dimension of the manifold $k$ and not the ambient dimension $d \gg k$.

**Corollary 4.** *Let $\varepsilon' \in (0, 1)$, $\eta > 0$, and let $\Omega \subset \mathbb{R}^d$ be a manifold of dimension $k \leqslant d$ satisfying Assumption 1. There exists $c_{\Omega, \eta} > 0$ not depending on $X$ or $n$ such that*

$$r^*(X, \eta, \varepsilon') \leqslant c_{\Omega, \eta} \left( \log \frac{n}{\varepsilon'} \right)^{5k/2}.$$

## 4 Sinkhorn Scaling an Approximate Kernel Matrix

The main result of this section, presented next, gives both a runtime bound and an error bound on the approximate Sinkhorn scaling performed in Algorithm 1. The error bound shows that the objective function $V_C(\cdot)$ in (1) is stable with respect to both (i) Sinkhorn projecting an approximate kernel matrix $\tilde{K}$ instead of the true kernel matrix $K$, and (ii) only performing an approximate Sinkhorn projection. The results of this section apply to any bounded cost matrix $C \in \mathbb{R}^{n \times n}$, with $\varepsilon' := \min(1, \frac{\varepsilon \eta}{50(\|C\|_\infty \eta + \log \frac{n}{\eta \varepsilon})})$.

**Theorem 5.** *If $K = e^{-\eta C}$ and if $\tilde{K} \in \mathbb{R}_{>0}^{n \times n}$ satisfies $\|\log K - \log \tilde{K}\|_\infty \leqslant \varepsilon'$, then the Sinkhorn subroutine in Algorithm 1 outputs $D_1, D_2$, and $\hat{W}$ such that $\tilde{P} := D_1 \tilde{K} D_2$ satisfies $\|\tilde{P}\mathbf{1} - \mathbf{p}\|_1 + \|\tilde{P}^\top \mathbf{1} - \mathbf{q}\|_1 \leqslant \varepsilon'$, $|V_C(P^\eta) - V_C(\tilde{P})| \leqslant \frac{\varepsilon}{2}$, and $|\hat{W} - V_C(\tilde{P})| \leqslant \frac{\varepsilon}{2}$. Moreover, if matrix-vector products can be computed with $\tilde{K}$ and $\tilde{K}^\top$ in time $\mathrm{T_{MULT}}$, then this takes time $\tilde{O}((n + \mathrm{T_{MULT}})\eta \|C\|_\infty \varepsilon'^{-1})$.*

The running time bound in Theorem 5 for the time required to produce $D_1$ and $D_2$ follows directly from prior work which has shown that Sinkhorn scaling can produce an approximation to the Sinkhorn projection of a positive matrix in time nearly independent of the dimension $n$ [3, 14]. The error bounds in Theorem 5 are based on Propositions 2 and 3.

**Proposition 2.** *For any $\mathbf{p}, \mathbf{q} \in \Delta_n$ and any $K, \tilde{K} \in \mathbb{R}_+^{n \times n}$,*

$$\|\Pi^{\mathcal{S}}(K) - \Pi^{\mathcal{S}}(\tilde{K})\|_1 \leqslant \|\log K - \log \tilde{K}\|_\infty.$$

**Proposition 3.** *Given $\tilde{K} \in \mathbb{R}_{>0}^{n \times n}$, let $\tilde{C} \in \mathbb{R}^{n \times n}$ satisfy $\tilde{C}_{ij} := -\eta^{-1} \log \tilde{K}_{ij}$. Let $D_1$ and $D_2$ be positive diagonal matrices such that $\tilde{P} := D_1 \tilde{K} D_2 \in \Delta_{n \times n}$, with $\delta := \|\mathbf{p} - \tilde{P}\mathbf{1}\|_1 + \|\mathbf{q} - \tilde{P}^\top \mathbf{1}\|_1$. If $\delta \leqslant 1$, then*

$$|V_{\tilde{C}}(\Pi^{\mathcal{S}}(\tilde{K})) - V_{\tilde{C}}(\tilde{P})| \leqslant \delta \|\tilde{C}\|_\infty + \eta^{-1} \delta \log \frac{2n}{\delta}.$$

## 5 Experimental Results

In this section we empirically validate our theoretical results. Details about the setup for each experiment appear in the supplement.

We first compare to the standard Sinkhorn algorithm. Fig. 1 plots the time-accuracy tradeoff for NYS-SINK, compared to the standard SINKHORN algorithm. Fig. 1 shows that NYS-SINK is consistently orders of magnitude faster to obtain the same accuracy.

We then investigate NYS-SINK's dependence on the intrinsic dimension and ambient dimension of the input. This is done by running NYS-SINK with a fixed approximation rank on distributions supported on 1-dimensional curves embedded in higher dimensions. Fig. 2 empirically validates the result in

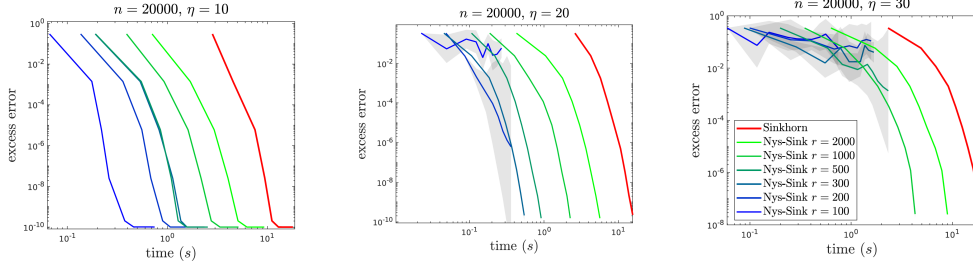

Figure 1: Time-accuracy tradeoff for NYS-SINK and SINKHORN, for a range of regularization parameters $\eta$ (each corresponding to a different Sinkhorn distance $W_\eta$) and approximation ranks $r$. Each experiment has been repeated 50 times; the variance is indicated by the shaded area around the curves. Note that curves in the plot start at different points corresponding to the time required for initialization.

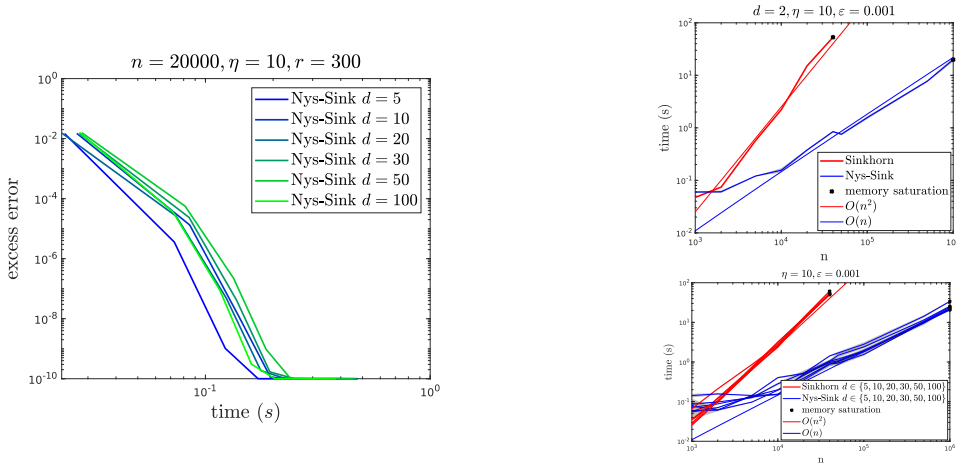

Figure 2: Accuracy of NYS-SINK as a function of running time, for different ambient dimensions.

Figure 3: Running time vs input size $n$ for NYS-SINK and SINKHORN. Top uses random point cloud data as in Fig. 1, bottom uses embedded curve data as in Fig. 2.

Corollary 4, namely that the required approximation rank – and consequently the computational complexity of NYS-SINK – is independent of the ambient dimension.

Next, we demonstrate NYS-SINK's dependence on the size $n$ of the dataset. As Fig. 3 indicates, the running time of NYS-SINK is empirically well-approximated by a line with slope 1 in the log-log plane – representing a complexity of $\Theta(n)$ – whereas the running time of SINKHORN scales as $\Theta(n^2)$.

Table 1: Performance of our algorithm on benchmark dataset.

| Exp. 1 $n = 3 \times 10^5, d = 3, \eta = 15$ | $W_2$ | time (s) |
|---|---|---|
| Nys-Sink $r = 2000, T = 20$ | $0.087 \pm 0.008$ | $0.4 \pm 0.1$ |
| Dual-Sink Multiscale + Anneal. $r = 0.95$ | 0.090 | 3.4 |
| Dual-Sink + Anneal. $r = 0.95$ | 0.087 | 35.4 |

| Exp. 2 $n = 3.8 \times 10^6, d = 3, \eta = 15$ | $W_2$ | time (s) |
|---|---|---|
| Nys-Sink $r = 2000, T = 20$ | $0.11 \pm 0.01$ | $6.3 \pm 0.8$ |
| Dual-Sink Multiscale + Anneal. $r = 0.95$ | 0.11 | 103.6 |
| Dual-Sink + Anneal. $r = 0.95$ | 0.10 | 1168 |

Moreover, SINKHORN saturates the RAM already for $n \approx 10^4$, whereas NYS-SINK can scale to $n \approx 10^6$ on the same machine.

Finally, we evaluate the performance of our algorithm on a benchmark dataset used in computer graphics. We measured Wasserstein distance between two 3D cloud points from the Stanford 3D Scanning Repository.[2] We ran two experiments, with $n = 3 \times 10^5$ and $n = 3.8 \times 10^6$ points, respectively.

We ran $T = 20$ iterations of our algorithm (Nys-Sink) with approximation rank $r = 2000$ on a GPU and compared to two optimized implementations in the library `GeomLoss`.[3] The results appear in Table 1. Each Nys-Sink experiment was repeated 50 times. Our method for moderate regularization $\eta$ is comparable with the other approaches in terms of precision, with a computational time that is orders of magnitude smaller. We note here that we choose the parameters $r, T$ in Nys-Sink by hand to balance precision and time complexity.

We note that in these experiments, instead of using our doubling-trick algorithm to choose the rank adaptively, we simply run experiments with a small fixed choice of $r$. As our experiments demonstrate, NYS-SINK achieves good empirical performance even when the rank $r$ is smaller than our theoretical analysis requires. Investigating this empirical success further is an interesting topic for future study.

## 6 Acknowledgments

We thank the reviewers for their helpful comments. We also thank Piotr Indyk, Pablo Parrilo, and Philippe Rigollet for helpful discussions. JA was supported in part by NSF Graduate Research Fellowship 1122374. FB and AR were supported in part by the European Research Council (grant SEQUOIA 724063). JNW was supported in part by the Josephine de Kármán Fellowship.

## Footnotes

[1] We use quotations since it is not technically a distance; see [13, Section 3.2] for details. The quotes are dropped henceforth.

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
