[Supplementary Material]

# Massively scalable Sinkhorn distances via the Nyström method: supplementary file

Jason Altschuler[*]
MIT
jasonalt@mit.edu

Francis Bach[†]
INRIA - ENS - PSL
francis.bach@inria.fr

Alessandro Rudi[†]
INRIA - ENS - PSL
alessandro.rudi@inria.fr

Jonathan Niles-Weed[‡]
NYU
jnw@cims.nyu.edu

**Abstract**

The Sinkhorn "distance," a variant of the Wasserstein distance with entropic regularization, is an increasingly popular tool in machine learning and statistical inference. However, the time and memory requirements of standard algorithms for computing this distance grow quadratically with the size of the data, making them prohibitively expensive on massive data sets. In this work, we show that this challenge is surprisingly easy to circumvent: combining two simple techniques—the Nyström method and Sinkhorn scaling—provably yields an accurate approximation of the Sinkhorn distance with significantly lower time and memory requirements than other approaches. We prove our results via new, explicit analyses of the Nyström method and of the stability properties of Sinkhorn scaling. We validate our claims experimentally by showing that our approach easily computes Sinkhorn distances on data sets hundreds of times larger than can be handled by other techniques.

[*]Supported in part by NSF Graduate Research Fellowship 1122374.
[†]Supported in part by the European Research Council (grant SEQUOIA 724063).
[‡]Supported in part by the Josephine de Kármán Fellowship.

# Contents

# 1    Introduction

Optimal transport is a fundamental notion in probability theory and geometry (Villani, 2008), which has recently attracted a great deal of interest in the machine learning community as a tool for image recognition (Li et al., 2013; Rubner et al., 2000), domain adaptation (Courty et al., 2014, 2017), and generative modeling (Arjovsky et al., 2017; Bousquet et al., 2017; Genevay et al., 2016), among many other applications (see, e.g., Kolouri et al., 2017; Peyré and Cuturi, 2017).

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

 (Williams and Seeger, 2001), sparse greedy approximations (Smola and Schölkopf, 2000), incomplete Cholesky decomposition (Fine and Scheinberg, 2001), Gram-Schmidt orthonormalization (Shawe-Taylor and Cristianini, 2004) or CUR matrix decompositions (Mahoney and Drineas, 2009). The approximation properties of these algorithms are now well understood (Alaoui and Mahoney, 2015; Bach, 2013; Gittens, 2011; Mahoney and Drineas, 2009); however, in this work, we require significantly more accurate bounds than are available from existing results as well as adaptive bounds for low-dimensional data. To establish these guarantees, we follow an approach based on approximation theory (see, e.g., Belkin, 2018; Rieger and Zwicknagl, 2010; Wendland, 2004), which consists of analyzing interpolation operators for the reproducing kernel Hilbert space corresponding to the Gaussian kernel.

Finally, this paper adds to recent work proposing the use of low-rank approximation for Sinkhorn scaling (Altschuler et al., 2018; Tenetov et al., 2018). We improve upon those papers in several ways. First, although we also exploit the idea of a low-rank approximation to the kernel matrix, we do so in a more sophisticated way that allows for automatic adaptivity to data with low-dimensional structure. These new approximation results are the key to our adaptive algorithm, and this yields a significant improvement in practice. Second, the analyses of Altschuler et al. (2018) and Tenetov et al. (2018) only yield an approximation to $W_\eta(\mathbf{p}, \mathbf{q})$ when $\eta \to \infty$. In the moderately regularized case when $\eta = O(1)$, which is typically used in practice, neither the work of Altschuler et al. (2018) nor of Tenetov et al. (2018) yields a rigorous error guarantee.

## 1.3  Outline of paper

Section 2 recalls preliminaries, and then formally states our main result and gives pseudocode for our proposed algorithm. The core of our theoretical analysis is in Sections 3 and 4. Section 3 presents our new results for Nyström approximation of Gaussian kernel matrices and Section 4 presents our new stability results for Sinkhorn scaling. Section 5 then puts these results together to conclude a proof for our main result (Theorem 1). Finally, Section 6 contains experimental results showing that our proposed algorithm outperforms state-of-the-art methods. The appendix contains proofs of several lemmas that are deferred for brevity of the main text.

## 2  Main result

### 2.1  Preliminaries and notation

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

$$r^*(X, \eta, \varepsilon') \leqslant (c(\eta R^2 + \log \tfrac{n}{\varepsilon'\eta}))^d.$$

*2. For any $k$-dimensional manifold $\Omega$ satisfying certain technical conditions and $\eta > 0$, there exists a constant $c_{\Omega,\eta}$ such that for any $n$ points lying on $\Omega$,*

$$r^*(X, \eta, \varepsilon') \leqslant c_{\Omega,\eta}(\log \tfrac{n}{\varepsilon'})^{5k/2}.$$

The formal versions of these bounds appear in Section 3. The second bound is significantly better than the first when $k \ll d$, and clearly shows the benefits of an adaptive procedure.

Combining Theorems 1 and 2 yields the following time and space complexity for our algorithm.

**Corollary 2** (Informal). *If $X$ consists of $n$ points lying in a ball of radius $R$ in $\mathbb{R}^d$, then with high probability Algorithm 1 requires*

$$\tilde{O}\left(n \cdot \frac{1}{\varepsilon}\left(c\eta R^2 + c \log \frac{n}{\varepsilon}\right)^{2d+1}\right) \text{ time and } \tilde{O}\left(n \cdot \left(c\eta R^2 + c \log \frac{n}{\varepsilon}\right)^d\right) \text{ space.}$$

*Moreover, if $X$ lies on a $k$-dimensional manifold $\Omega$, then with high probability Algorithm 1 requires*

$$\tilde{O}\left(n \cdot \frac{c_{\Omega,\eta}}{\varepsilon}\left(\log \frac{n}{\varepsilon}\right)^{5k}\right) \text{ time and } \tilde{O}\left(n \cdot c_{\Omega,\eta}\left(\log \frac{n}{\varepsilon}\right)^{5k/2}\right) \text{ space.}$$

Altschuler et al. (2017) noted that an approximation to the *unregularized* optimal transport cost is obtained by taking $\eta = \Theta\left(\varepsilon^{-1}\log n\right)$. Thus it follows that Algorithm 1 computes an additive $\varepsilon$ approximation to the unregularized transport distance in $O\left(n\left(\varepsilon^{-1}R^2\log n\right)^{O(d)}\right)$ time with high probability. However, a theoretically better running time for that problem can be obtained by a simple but impractical algorithm based on rounding the input distributions to an $\varepsilon$-net and then running Sinkhorn scaling on the resulting instance.[2]

## 3 Kernel approximation via the Nyström method

In this section, we describe the algorithm ADAPTIVENYSTRÖM used in line 2 of Algorithm 1 and bound its runtime complexity, space complexity, and error. We first establish basic properties of Nyström approximation and give pseudocode for ADAPTIVENYSTRÖM (Sections 3.1 and 3.2) before stating and proving formal versions of the bounds appearing in Theorem 2 (Sections 3.3 and 3.4).

### 3.1 Preliminaries: Nyström and error in terms of effective dimension

Given points $X = \{x_1, \ldots, x_n\}$ with $\|x_i\|_2 \leqslant R$ for all $i \in [n]$, let $K \in \mathbb{R}^{n \times n}$ denote the matrix with entries $K_{ij} := k_\eta(x_i, x_j)$, where $k_\eta(x, x') := e^{-\eta\|x-x'\|^2}$. Note that $k_\eta(x, x')$ is the Gaussian kernel $e^{-\|x-x'\|^2/(2\sigma^2)}$ between points $x$ and $x'$ with bandwith parameter $\sigma^2 = \frac{1}{2\eta}$. For $r \in \mathbb{N}$, we consider an approximation of the matrix $K$ that is of the form

$$\widetilde{K} = VA^{-1}V^\top,$$

where $V \in \mathbb{R}^{n \times r}$ and $A \in \mathbb{R}^{r \times r}$. In particular we will consider the approximation given by the Nyström method which, given a set $X_r = \{\widetilde{x}_1, \ldots, \widetilde{x}_r\} \subset X$, constructs $V$ and $A$ as:

$$V_{ij} = k_\eta(x_i, \widetilde{x}_j), \quad A_{jj'} = k_\eta(\widetilde{x}_j, \widetilde{x}'_j),$$

for $i \in [n]$ and $j, j' \in [r]$. Note that the matrix $\widetilde{K}$ is never computed explicitly. Indeed, our proposed Algorithm 1 only depends on $\widetilde{K}$ through computing matrix-vector products $\widetilde{K}v$, where $v \in \mathbb{R}^n$, and these can be computed efficiently as

$$\widetilde{K}v = V(L^{-\top}(L^{-1}(V^\top v))), \tag{4}$$

where $L \in \mathbb{R}^{r \times r}$ is the lower triangular matrix satisfying $LL^\top = A$ obtained by the Cholesky decomposition of $A$, and where we compute products of the form $L^{-1}v$ (resp. $L^{-\top}v$) by solving the triangular system $Lx = v$ (resp. $L^\top x = v$). Once a Cholesky decomposition of $A$ has been obtained—at computational cost $O(r^3)$—matrix-vector products can therefore be computed in time $O(nr)$.

We now turn to understanding the approximation error of this method. In this paper we will sample the set $X_r$ via *approximate leverage-score sampling*. In particular, we do this via Algorithm 2 of Musco and Musco (2017). The following lemma shows that taking the rank $r$ to be on the order of the effective dimension $d_{\text{eff}}(\tau)$ (see Definition 3) is sufficient to guarantee that $\widetilde{K}$ approximates $K$ to within error $\tau$ in operator norm.

**Lemma 1.** *Let $\tau, \delta > 0$. Consider sampling $X_r$ from $X$ according to Algorithm 2 of Musco and Musco (2017), for some positive integer $r \geqslant 400 d_{eff}(\tau) \log \frac{3n}{\delta}$. Then:*

1. *Sampling $X_r$ and forming the matrices $V$ and $L$ (which define $\tilde{K}$, see (4)) requires $O(nr^2 + r^3)$ time and $O(n(r+d))$ space.*

2. *Computing matrix-vector products with $\tilde{K}$ can be done in time $O(nr)$.*

3. *With probability at least $1 - \delta$, $\|K - \widetilde{K}\|_{\mathrm{op}} \leqslant \tau n$.*

*Proof.* The result follows directly from Theorem 7 of Musco and Musco (2017) and the fact that $d_{\mathrm{eff}}(\tau) \leqslant \mathrm{rank}(K) \leqslant n$ for any $\tau \geqslant 0$. □

## 3.2 Adaptive Nyström with doubling trick

Here we give pseudocode for the ADAPTIVENYSTRÖM subroutine in Algorithm 2. The algorithm is based on a simple doubling trick, so that the rank of the approximate kernel can be chosen adaptively. The observation enabling this trick is that given a Nyström approximation $\tilde{K}$ to the actual kernel matrix $K = e^{-\eta \|x_i - x_j\|_2^2}$, the entrywise error $\|K - \tilde{K}\|_\infty$ of the approximation can be computed exactly in $O(nr^2)$ time. The reason for this is that (i) the entrywise norm $\|K - \tilde{K}\|_{\max}$ is equal to the maximum entrywise error on the diagonal $\max_{i \in [n]} |K_{ii} - \tilde{K}_{ii}| = 1 - \min_{i \in [n]} \tilde{K}_{ii}$, proven below in Lemma 2; and (ii) the quantity $1 - \min_{i \in [n]} \tilde{K}_{ii}$ is easy to compute quickly.

Below, line 4 in Algorithm 2 denotes the approximate leverage-score sampling scheme of Musco and Musco (2017, Algorithm 2) when applied to the Gaussian kernel matrix $K_{ij} := e^{-\eta \|x_i - x_j\|^2}$. We note that the BLESS algorithm of Rudi et al. (2018) allows for re-using previously sampled points when doubling the sampling rank. Although this does not affect the asymptotic runtime, it may lead to speedups in practice.

---

**Input:** $X = \{x_1, \ldots, x_n\} \in \mathbb{R}^{d \times n}$, $\eta > 0$, $\tau > 0$
**Output:** $\tilde{K} \in \mathbb{R}^{n \times n}$, $r \in \mathbb{N}$
 1: err $\leftarrow +\infty$, $r \leftarrow 1$
 2: **while** err $> \tau$ **do**
 3:      $r \leftarrow 2r$
 4:      $\tilde{K} \leftarrow$ NYSTRÖM$(X, \eta, r)$
 5:      err $\leftarrow 1 - \min_{i \in [n]} \tilde{K}_{ii}$
 6: **end while**
 7: **return** $(\tilde{K}, \mathrm{rank}(\tilde{K}))$

---

**Algorithm 2:** ADAPTIVENYSTRÖM

**Lemma 2.** *Let $(\tilde{K}, r)$ denote the (random) output of ADAPTIVENYSTRÖM$(X, \eta, \tau)$. Then:*

1. *$\|K - \tilde{K}\|_\infty \leqslant \tau$.*

2. *The algorithm used $O(nr)$ space and terminated in $O(nr^2)$ time.*

3. *There exists a universal constant c such that simultaneously for every $\delta > 0$,*

$$\mathbb{P}\left(r \leqslant c \cdot d_{\text{eff}}\left(\tfrac{\tau}{n}\right) \log\left(\tfrac{n}{\delta}\right)\right) \geqslant 1 - \delta.$$

*Proof.* By construction, the Nyström approximation $\tilde{K}$ is a PSD approximation of $K$ in the sense that $K \succeq \tilde{K} \succeq 0$, see e.g., Musco and Musco (2017, Theorem 3). Since Sylvester's criterion for $2 \times 2$ minors guarantees that the maximum modulus entry of a PSD matrix is always achieved on the diagonal, it follows that $\|K - \tilde{K}\|_\infty = \max_{i \in [n]} |K_{ii} - \tilde{K}_{ii}|$. Now each $K_{ii} = 1$ by definition of $K$, and each $\tilde{K}_{ii} \in [0, 1]$ since $K \succeq \tilde{K} \succeq 0$. Therefore we conclude

$$\|K - \tilde{K}\|_\infty = 1 - \min_{i \in [n]} \tilde{K}_{ii}.$$

This implies Item 1. Item 2 follows upon using the space and runtime complexity bounds in Lemma 1 and noting that the final call to Nyström is the dominant for both space and runtime. Item 3 is immediate from Lemma 1 and the fact that $\|K - \tilde{K}\|_\infty \leqslant \|K - \tilde{K}\|_{\text{op}}$ (Lemma J). $\qquad\square$

## 3.3 General results: data points lie in a ball

In this section we assume no structure on $X$ apart from the fact that $X \subseteq B_R^d$ where $B_R^d$ is a ball of radius $R$ in $\mathbb{R}^d$ centered around the origin, for some $R > 0$ and $d \in \mathbb{N}$. First we characterize the eigenvalues of $K$ in terms of $\eta, d, R$, and then we use this to bound $d_{\text{eff}}$.

**Theorem 3.** *Let $X := \{x_1, \ldots x_n\} \subseteq B_R^d$, and let $K \in \mathbb{R}^{n \times n}$ be the matrix with entries $K_{ij} := e^{-\eta\|x_i - x_j\|^2}$. Then:*

1. *For each $t \in \mathbb{N}, t \geqslant (2e)^d$, $\lambda_{t+1}(K) \leqslant n e^{-\frac{d}{2e} t^{1/d} \log \frac{d\, t^{1/d}}{4e^2 \eta R^2}}$.*

2. *For each $\tau \in (0, 1]$, $d_{\text{eff}}(\tau) \leqslant 3\left(6 + \frac{41}{d}\eta R^2 + \frac{3}{d}\log\frac{1}{\tau}\right)^d$.*

We sketch the proof of Theorem 3 here; details are deferred to Appendix B.5 for brevity of the main text. We begin by recalling the argument of Cotter et al. (2011) that truncating the Taylor expansion of the Gaussian kernel guarantees for each positive integer $T$ the existence of a rank $M_T := \binom{d+T}{T}$ matrix $\tilde{K}_T$ satisfying

$$\|K - \tilde{K}_T\|_\infty \leqslant \frac{(2\eta R^2)^{T+1}}{(T+1)!}.$$

On the other hand, by the Eckart-Young-Mirsky Theorem,

$$\lambda_{M_T+1} = \inf_{\bar{K}_T \in \mathbb{R}^{n \times n},\, \text{rank}(\bar{K}_T) \leqslant M_T} \|K - \bar{K}_T\|_{\text{op}}.$$

Therefore by combining the above two displays, we conclude that

$$\lambda_{M_T+1} \leqslant \|K - \tilde{K}_T\|_{\text{op}} \leqslant n\|K - \tilde{K}_T\|_\infty \leqslant n\frac{(2\eta R^2)^{T+1}}{(T+1)!}.$$

Proofs of the two claims follow by bounding this quantity. Details are in Appendix B.5.

Theorem 3 characterizes the eigenvalue decay and effective dimension of Gaussian kernel matrices in terms of the dimensionality of the space, with explicit constants and explicit dependence on the width parameter $\eta$ and the radius $R$ of the ball (see Belkin, 2018, for asymptotic results). This yields the following bound on the optimal rank for approximating Gaussian kernel matrices of data lying in a Euclidean ball.

**Corollary 3.** *Let $\varepsilon' \in (0,1)$ and $\eta > 0$. If $X$ consists of $n$ points lying in a ball of radius $R$ around the origin in $\mathbb{R}^d$, then*

$$r^*(X, \eta, \varepsilon') \leqslant 3 \left( 6 + \frac{53}{d}\eta R^2 + \frac{3}{d} \log \frac{2n}{\varepsilon'} \right)^d$$

*Proof.* Directly from the explicit bound of Theorem 3 and the definition of $r^*(X, \eta, \varepsilon')$. $\square$

## 3.4 Adaptivity: data points lie on a low dimensional manifold

In this section we consider $X \subset \Omega \subset \mathbb{R}^d$, where $\Omega$ is a low dimensional manifold. In Theorem 4 we give a result about the approximation properties of the Gaussian kernel over manifolds and a bound on the eigenvalue decay and effective dimension of Gaussian kernel matrices. We prove that the effective dimension is logarithmic in the precision parameter $\tau$ to a power depending only on the dimensionality $k$ of the manifold (to be contrasted to the dimensionality of the ambient space $d \gg k$).

Let $\Omega \subset \mathbb{R}^d$ be a smooth compact manifold without boundary, and $k < d$. Let $(\Psi_j, U_j)_{j \in [T]}$, with $T \in \mathbb{N}$, be an atlas for $\Omega$, where without loss of generality, $(U_j)_j$ are open sets covering $\Omega$, $\Psi_j : U_j \to B_{r_j}^k$ are smooth maps with smooth inverses, mapping $U_j$ bijectively to $B_{r_j}^k$, balls of radius $r_j$ centered around the origin of $\mathbb{R}^k$. We assume the following quantitative control on the smoothness of the atlas.

**Assumption 1.** There exists $Q > 0$ such that

$$\sup_{u \in B_{r_j}^k} \|D^\alpha \Psi_j^{-1}(u)\| \leqslant Q^{|\alpha|}, \qquad \alpha \in \mathbb{N}^k, j \in [T],$$

where $|\alpha| = \sum_{j=1}^k \alpha_j$ and $D^\alpha = \frac{\partial^{|\alpha|}}{\partial u_1^{\alpha_1} \dots \partial u_k^{\alpha_k}}$, for $\alpha \in \mathbb{N}^k$.

Before stating our result, we need to introduce the following helpful definition. Given $f : \mathbb{R}^d \to \mathbb{R}$, and $X = \{x_1, \dots, x_n\} \subset \Omega$, denote by $\widehat{f}_X$ the function

$$\widehat{f}_X(x) := \sum_{i=1}^n c_i k_\eta(x, x_i), \quad c = K^{-1} v_f,$$

with $v_f = (f(x_1), \dots, f(x_n))$ and $K \in \mathbb{R}^{n \times n}$ the kernel matrix over $X$, i.e. $K_{ij} = k_\eta(x_i, x_j)$. Note that $\widehat{f}_X(x_i) = f(x_i)$ by construction (Wendland, 2004). We have the following result.

**Theorem 4.** *Let $\Omega \subset B_R^d \subset \mathbb{R}^d$ be a smooth compact manifold without boundary satisfying Assumption 1. Let $X \subset \Omega$ be a set of cardinality $n \in \mathbb{N}$. Then the following holds*

1. Let $h_{X,\Omega} = \sup_{x' \in \Omega} \inf_{x \in X} \|x - x'\|$. Let $\mathcal{H}$ be the RKHS associated to the Gaussian kernel of a given width. There exist $c, h > 0$ not depending on $X, n$, such that, when $h_{X,\Omega} \leqslant h$ the following holds

$$|f(x) - \widehat{f}_X(x)| \leqslant e^{-ch_{X,\Omega}^{-2/5}} \|f\|_{\mathcal{H}}, \qquad \forall f \in \mathcal{H}, \quad x \in \Omega.$$

2. Let $K$ be the Gaussian kernel matrix associated to $X$. Then there exists a constant $c$ not depending on $X$ or $n$, for which

$$\lambda_{p+1}(K) \leqslant n e^{-cp^{\frac{2}{5k}}}, \qquad \forall p \in [n].$$

3. Let $\tau \in (0, 1]$. Let $K$ be the Gaussian kernel matrix associated to $X$ and $d_{eff}(\tau)$ the effective dimension computed on $K$. There exists $c_1, c_2$ not depending on $X$, $n$, or $\tau$, for which

$$d_{eff}(\tau) \leqslant \left( c_1 \log \frac{1}{\tau} \right)^{5k/2} + c_2.$$

*Proof.* First we recall some basic multi-index notation and introduce Sobolev Spaces. When $\alpha \in \mathbb{N}_0^d, x \in \mathbb{R}^d, g : \mathbb{R}^d \to \mathbb{R}$, we write

$$x^\alpha = \prod_i x_i^{\alpha_i}, \quad |\alpha| = \sum_i \alpha_i, \quad \alpha! = \prod_i \alpha_i!, \quad D^\alpha = \frac{\partial^{|\alpha|}}{\partial x_1^{\alpha_1} \dots \partial x_n^{\alpha_n}}.$$

Next, we recall the definition of Sobolev spaces. For $m, p \in \mathbb{N}$ and $B \subseteq \mathbb{R}^k$, define the norm $\| \cdot \|_{W_p^m(B)}$ by

$$\|f\|_{W_p^m(B)}^p = \sum_{|\alpha| \leqslant m} \|D^\alpha f\|_{L^p(B)}^p,$$

and the space of $W_p^m(B)$ as $W_p^m(B) = \overline{C^\infty(B)}^{\|\cdot\|_{W_p^m(B)}}$.

For any $j \in [T]$, $u \in \mathcal{H}$ we have the following. By Lemma O, we have that there exists a constant $C_{d,k,R,r_j}$ such that for any $q \geqslant k$,

$$\|u \circ \Psi_j^{-1}\|_{W_2^q(B_{r_j}^k)} \leqslant C_{d,k,R,r_j} q^k (2qdQ)^q \|u\|_{W_2^{q+(d+1)/2}(B_R^d)}.$$

Now note that by Theorem 7.5 of Rieger and Zwicknagl (2010) we have that there exists a constant $C_\eta$ such that

$$\|u\|_{W_2^{q+(d+1)/2}(B_R^d)} \leqslant \|u\|_{W_2^{q+(d+1)/2}(\mathbb{R}^d)} \leqslant (C_\eta)^{q+(d+1)/2} \left( q + \frac{d+1}{2} \right)^{\frac{q}{2} + \frac{d+1}{4}} \|u\|_{\mathcal{H}}.$$

Then, since $q^m \leqslant m^m (1+m)^q$, for any $q \geqslant 1$, we have

$$\|u \circ \Psi_j^{-1}\|_{W_2^q(B_{r_j}^k)} \leqslant C_{d,k,R,r_j,Q,\eta}^q q^{\frac{3q}{2}} \|u\|_{\mathcal{H}},$$

for a suitable constant $C_{d,k,R,r_j,Q,\eta}$ depending on $C_{d,k,R,r_j,Q}$, $C_\eta$ and $(d+1)/2$.

In particular we want to study $\|u\|_{L^\infty(\Omega)}$, for $u = f - \widehat{f}_X$. We have

$$\|u\|_{L^\infty(\Omega)} = \sup_{j \in [T]} \|u\|_{L^\infty(U_j)} = \sup_{j \in [T]} \|u \circ \Psi_j^{-1}\|_{L^\infty(B_{r_j}^k)}.$$

Now for $j \in [T]$, denote by $Z_j$ the set $Z_j = \{\Psi_j(x) | x \in X \cap U_j\}$. By construction of $u = f - \widehat{f}_X$, we have

$$(u \circ \Psi_j^{-1})|_{Z_j} = u|_{X \cap U_j} = 0.$$

Define $h_{Z_j, B_{r_j}^k} = \sup_{z \in B_{r_j}^k} \inf_{z' \in Z_j} \|z - z'\|$. We have established that there exists $C > 0$, such that $\|u \circ \Psi_j^{-1}\|_{W_2^q(B_{r_j}^k)} \leqslant C^q q^{\frac{3}{2}q} \|u\|_{\mathcal{H}}$, and by construction $(u \circ \Psi_j)|_{Z_j} = 0$. We can therefore apply Theorem 3.5 of Rieger and Zwicknagl (2010) to obtain that there exists a $c_j, h_j > 0$, for which, when $h_{Z_j, B_{r_j}^k} \leqslant h_j$, then

$$\|u \circ \Psi_j^{-1}\|_{L^\infty(B_{r_j}^k)} \leqslant \exp\left(-c_j h_{Z_j, B_{r_j}^k}^{-2/5}\right) \|u\|_{\mathcal{H}}.$$

Now, denote by $\bar{h}_{S,U} = \sup_{x' \in U} \inf_{x \in S} d(x, x')$ with $d$ the geodesic distance over the manifold $\Omega$. By applying Theorem 8 of Fuselier and Wright (2012), we have that there exist $C$ and $h_0$ not depending on $X$ or $n$ such that, when $\bar{h}_{X,\Omega} \leqslant h_0$, the inequality $\bar{h}_{X_j \cap U_j, U_j} \leqslant C \bar{h}_{X,\Omega}$ holds for any $j \in [T]$. Moreover, since by Theorem 6 of the same paper $\|x - x'\| \leqslant d(x, x') \leqslant C_1 \|x - x'\|$, for $C_1 > 1$ and $x, x' \in \Omega$, then

$$h_{Z_j, B_{r_j}^k} \leqslant \bar{h}_{X_j \cap U_j, U_j} \leqslant C \bar{h}_{X,\Omega} \leqslant C C_1 h_{X,\Omega}.$$

Finally, defining $c_1 = c(2 \max_j C_j)^{-2/5}, h = C_1^{-1} \min(h_0, C^{-1} \min_j h_j)$, when $h_{X,\Omega} \leqslant h$,

$$\|f - \widehat{f}_X\|_{L^\infty(\Omega)} = \max_{j \in [T]} \|u \circ \Psi_j^{-1}\|_{L^\infty(B_{r_j}^k)} \leqslant e^{-c_1 h_{X,\Omega}^{-2/5}} \|f\|_{\mathcal{H}}, \qquad \forall f \in \mathcal{H}, \ \ x \in \Omega.$$

The proof of Points 2 and 3 now proceeds as in Theorem 3. Details are deferred to Appendix B.5. $\qquad\square$

Point 1 of the result above is new, to our knowledge, and extends interpolation results on manifolds (Fuselier and Wright, 2012; Hangelbroek et al., 2010; Wendland, 2004), from polynomial to exponential decay, generalizing a technique of Rieger and Zwicknagl (2010) to a subset of real analytic manifolds. Points 2 and 3 are a generalization of Theorem 3 to the case of manifolds. In particular, the crucial point is that now the eigenvalue decay and the effective dimension depend on the dimension of the manifold $k$ and not the ambient dimension $d \gg k$. We think that the factor $5/2$ in the exponent of the eigenvalues and effective dimension is a result of the specific proof technique used and could be removed with a refined analysis, which is out of the scope of this paper.

We finally conclude the desired bound on the optimal rank in the manifold case.

**Corollary 4.** *Let $\varepsilon' \in (0,1)$, $\eta > 0$, and let $\Omega \subset \mathbb{R}^d$ be a manifold of dimensionality $k \leqslant d$ satisfying Assumption 1. There exists $c_{\Omega,\eta} > 0$ not depending on $X$ or $n$ such that*

$$r^*(X, \eta, \varepsilon') \leqslant c_{\Omega,\eta} \left(\log \frac{n}{\varepsilon'}\right)^{5k/2}$$

*Proof.* By the definition of $r^*(X, \eta, \varepsilon')$ and the bound of Theorem 4, we have

$$r^*(X, \eta, \varepsilon') \leqslant \left(c_1(4\eta R^2 + \log \tfrac{2n}{\varepsilon'})\right)^{5k/2} + c_2 \,.$$

Since $\log \tfrac{2n}{\varepsilon'} \geqslant 1$, we may set $c_{\Omega,\eta} = \max\left\{(8c_1 \eta R^2)^{5k/2}, c_2\right\}$ to obtain the claim. $\qquad\square$

# 4  Sinkhorn scaling an approximate kernel matrix

The main result of this section, presented next, gives both a runtime bound and an error bound on the approximate Sinkhorn scaling performed in line 3 of Algorithm 1.[3]  The runtime bound shows that we only need a small number of iterations to perform this approximate Sinkhorn projection on the approximate kernel matrix.  The error bound shows that the objective function $V_C(\cdot)$ in (1) is stable with respect to both (i) Sinkhorn projecting an *approximate kernel matrix* $\tilde{K}$ instead of the true kernel matrix $K$, and (ii) only performing an *approximate Sinkhorn projection*.

The results of this section apply to any bounded cost matrix $C \in \mathbb{R}^{n \times n}$, not just the cost matrix $C_{ij} = \|x_i - x_j\|_2^2$ for the squared Euclidean distance. To emphasize this, we state this result and the rest of this section in terms of an arbitrary such matrix $C$. Note that $\|C\|_\infty \leqslant 4R^2$ when $C_{ij} = \|x_i - x_j\|_2^2$ and all points lie in a Euclidean ball of radius $R$. We therefore state all results in this section for $\varepsilon' := \min(1, \frac{\varepsilon\eta}{50(\|C\|_\infty \eta + \log \frac{n}{\eta\varepsilon})})$.

**Theorem 5.** *If $K = e^{-\eta C}$ and if $\tilde{K} \in \mathbb{R}_{>0}^{n \times n}$ satisfies $\|\log K - \log \tilde{K}\|_\infty \leqslant \varepsilon'$, then Line 3 of Algorithm 1 outputs $D_1$, $D_2$, and $\hat{W}$ such that $\tilde{P} := D_1 \tilde{K} D_2$ satisfies $\|\tilde{P}\mathbf{1} - \mathbf{p}\|_1 + \|\tilde{P}^\top \mathbf{1} - \mathbf{q}\|_1 \leqslant \varepsilon'$ and*

$$|V_C(P^\eta) - V_C(\tilde{P})| \leqslant \frac{\varepsilon}{2} \tag{5a}$$

$$|\hat{W} - V_C(\tilde{P})| \leqslant \frac{\varepsilon}{2} \tag{5b}$$

*Moreover, if matrix-vector products can be computed with $\tilde{K}$ and $\tilde{K}^\top$ in time $\mathrm{T_{MULT}}$, then this takes time $\tilde{O}((n + \mathrm{T_{MULT}})\eta\|C\|_\infty \varepsilon'^{-1})$.*

The running time bound in Theorem 5 for the time required to produce $D_1$ and $D_2$ follows directly from prior work which has shown that Sinkhorn scaling can produce an approximation to the Sinkhorn projection of a positive matrix in time nearly independent of the dimension $n$.

**Theorem 6** (Altschuler et al., 2017; Dvurechensky et al., 2018)**.** *Given a matrix $\tilde{K} \in \mathbb{R}_{>0}^{n \times n}$, the Sinkhorn scaling algorithm computes diagonal matrices $D_1$ and $D_2$ such that $\tilde{P} := D_1 \tilde{K} D_2$ satisfies $\|\tilde{P}\mathbf{1} - \mathbf{p}\|_1 + \|\tilde{P}^\top \mathbf{1} - \mathbf{q}\|_1 \leqslant \delta$ in $O(\delta^{-1} \log \frac{n}{\delta \min_{ij} \tilde{K}_{ij}})$ iterations, each of which requires $O(1)$ matrix-vector products with $\tilde{K}$ and $O(n)$ additional processing time.*

Lemma A establishes that computing the approximate cost $\hat{W}$ requires $O(n + \mathrm{T_{MULT}})$ additional time. To obtain the running time claimed in Theorem 5, it therefore suffices to use the fact that $\log \frac{1}{\min_{ij} \tilde{K}_{ij}} \leqslant \log \frac{1}{\min_{ij} K_{ij}} + \| \log K - \log \tilde{K} \|_\infty \leqslant \eta \| C \|_\infty + \varepsilon'$.

The remainder of the section is devoted to proving the error bounds in Theorem 5. Subsection 4.1 proves stability bounds for using an approximate kernel matrix, Subsection 4.2 proves stability bounds for using an approximate Sinkhorn projection, and then Subsection 4.3 combines these results to prove the error bounds in Theorem 5.

## 4.1   Using an approximate kernel matrix

Here we present the first ingredient for the proof of Theorem 5: that Sinkhorn projection is Lipschitz with respect to the logarithm of the matrix to be scaled. If we view Sinkhorn projection as a saddle-point approximation to a Gibbs distribution over the vertices of $\mathcal{M}(\mathbf{p}, \mathbf{q})$ (see discussion by Kosowsky and Yuille, 1994a), then this result is analogous to the fact that the total variation between Gibbs distributions is controlled by the $\ell_\infty$ distance between the energy functions (Simon, 1979).

**Proposition 2.** *For any* $\mathbf{p}, \mathbf{q} \in \Delta_n$ *and any* $K, \tilde{K} \in \mathbb{R}_+^{n \times n}$,

$$\| \Pi^{\mathcal{S}}(K) - \Pi^{\mathcal{S}}(\tilde{K}) \|_1 \leqslant \| \log K - \log \tilde{K} \|_\infty \,.$$

*Proof.* Note that $-H(P)$ is 1-strongly convex with respect to the $\ell_1$ norm (Bubeck, 2015, Section 4.3). By Proposition 1, $\Pi^{\mathcal{S}}(K) = \mathrm{argmin}_{P \in \mathcal{M}(\mathbf{p}, \mathbf{q})} \langle -\log K, P \rangle - H(P)$ and $\Pi^{\mathcal{S}}(\tilde{K}) = \mathrm{argmin}_{P \in \mathcal{M}(\mathbf{p}, \mathbf{q})} \langle -\log \tilde{K}, P \rangle - H(P)$. The claim follows upon applying Lemma I.  □

In words, Proposition 2 establishes that the Sinkhorn projection operator is Lipschitz on the "logarithmic scale." By contrast, we show in Appendix C that the Sinkhorn projection does not satisfy a Lipschitz property in the standard sense for any choice of matrix norm.

## 4.2   Using an approximate Sinkhorn projection

Here we present the second ingredient for the proof of Theorem 5: that the objective function $V_C(\cdot)$ for Sinkhorn distances in (1) is stable with respect to the target row and column sums $\mathbf{p}$ and $\mathbf{q}$ of the outputted matrix.

**Proposition 3.** *Given* $\tilde{K} \in \mathbb{R}_{>0}^{n \times n}$, *let* $\tilde{C} \in \mathbb{R}^{n \times n}$ *satisfy* $\tilde{C}_{ij} := -\eta^{-1} \log \tilde{K}_{ij}$. *Let* $D_1$ *and* $D_2$ *be positive diagonal matrices such that* $\tilde{P} := D_1 \tilde{K} D_2 \in \Delta_{n \times n}$, *with* $\delta := \| \mathbf{p} - \tilde{P} \mathbf{1} \|_1 + \| \mathbf{q} - \tilde{P}^\top \mathbf{1} \|_1$. *If* $\delta \leqslant 1$, *then*

$$|V_{\tilde{C}}(\Pi^{\mathcal{S}}(\tilde{K})) - V_{\tilde{C}}(\tilde{P})| \leqslant \delta \| \tilde{C} \|_\infty + \eta^{-1} \delta \log \frac{2n}{\delta} \,,$$

*Proof.* Write $\tilde{\mathbf{p}} := \tilde{P} \mathbf{1}$ and $\tilde{\mathbf{q}} := \tilde{P}^\top \mathbf{1}$. Then $\tilde{P} = \Pi^{\mathcal{S}}_{\mathcal{M}(\tilde{\mathbf{p}}, \tilde{\mathbf{q}})}(\tilde{K})$ by the definition of the

Sinkhorn projection. If we write $P^* := \Pi^{\mathcal{S}}_{\mathcal{M}(\mathbf{p},\mathbf{q})}(\tilde{K})$, then Proposition 1 implies

$$\tilde{P} = \underset{P \in \mathcal{M}(\tilde{\mathbf{p}},\tilde{\mathbf{q}})}{\operatorname{argmin}} V_{\tilde{C}}(P)$$

$$P^* = \underset{P \in \mathcal{M}(\mathbf{p},\mathbf{q})}{\operatorname{argmin}} V_{\tilde{C}}(P)$$

Lemma G establishes that the Hausdorff distance between $\mathcal{M}(\tilde{\mathbf{p}},\tilde{\mathbf{q}})$ and $\mathcal{M}(\mathbf{p},\mathbf{q})$ with respect to $\|\cdot\|_1$ is at most $\delta$, and by Lemma E, the function $V_{\tilde{C}}$ satisfies

$$|V_{\tilde{C}}(P) - V_{\tilde{C}}(Q)| \leqslant \omega(\|P - Q\|_1) \,,$$

where $\omega(\delta) := \delta \|\tilde{C}\|_\infty + \eta^{-1}\delta \log \frac{2n}{\delta}$ is increasing and and continuous on $[0,1]$ as long as $n \geqslant 2$. Applying Lemma H then yields the claim. $\qquad\square$

### 4.3 Proof of Theorem 5

The runtime claim was proven in Section 4; here we prove the error bounds. We first show (5a). Define $\tilde{C} := -\eta^{-1}\log\tilde{K}$. Since $P^\eta = \Pi^{\mathcal{S}}(K)$ by Corollary 1, we can decompose the error as

$$|V_C(P^\eta) - V_C(\tilde{P})| \leqslant \left| V_C\left(\Pi^{\mathcal{S}}(K)\right) - V_C\left(\Pi^{\mathcal{S}}\left(\tilde{K}\right)\right) \right| \tag{6a}$$

$$+ \left| V_C\left(\Pi^{\mathcal{S}}\left(\tilde{K}\right)\right) - V_{\tilde{C}}\left(\Pi^{\mathcal{S}}\left(\tilde{K}\right)\right) \right| \tag{6b}$$

$$+ \left| V_{\tilde{C}}\left(\Pi^{\mathcal{S}}\left(\tilde{K}\right)\right) - V_{\tilde{C}}\left(\tilde{P}\right) \right| \tag{6c}$$

$$+ \left| V_{\tilde{C}}\left(\tilde{P}\right) - V_C\left(\tilde{P}\right) \right| . \tag{6d}$$

By Proposition 2 and Lemma E, term (6a) is at most $\varepsilon'\|C\|_\infty + \eta^{-1}\varepsilon'\log\frac{2n}{\varepsilon'}$. Proposition 3 implies that (6c) is at most $\varepsilon'\|\tilde{C}\|_\infty + \eta^{-1}\varepsilon'\log\frac{2n}{\varepsilon'}$. Finally, by Lemma C, terms (6b) and (6d) are each at most $\eta^{-1}\varepsilon'$. Thus

$$|V_C(\Pi^{\mathcal{S}}(K)) - V_C(\tilde{P})| \leqslant \left(\varepsilon'\|C\|_\infty + \eta^{-1}\varepsilon'\log\frac{2n}{\varepsilon'}\right) + \left(\varepsilon'\|\tilde{C}\|_\infty + \eta^{-1}\varepsilon'\log\frac{2n}{\varepsilon'}\right) + 2\eta^{-1}\varepsilon'$$

$$\leqslant 2\varepsilon'\|C\|_\infty + \eta^{-1}(\varepsilon'^2 + 2\varepsilon') + 2\eta^{-1}\varepsilon'\log\frac{2n}{\varepsilon'}$$

$$\leqslant \varepsilon'(2\|C\|_\infty + 3\eta^{-1}) + 2\eta^{-1}\varepsilon'\log\frac{2n}{\varepsilon'} \,,$$

where the second inequality follows from the fact that $\|\tilde{C}\|_\infty \leqslant \|C\|_\infty + \|C - \tilde{C}\|_\infty \leqslant \|C\|_\infty + \eta^{-1}\varepsilon'$. The proof of (5a) is then complete by invoking Lemma M.

To prove (5b), by Lemma A we have $\hat{W} = V_{\tilde{C}}(\tilde{P})$, and by Lemma C, we therefore have $|\hat{W} - V_C(\tilde{P})| \leqslant \eta^{-1}\varepsilon' \leqslant \frac{\varepsilon}{2}$.

## 5 Proof of Theorem 1

In this section, we combine the results of the preceding three sections to prove Theorem 1.

**Error analysis.** First, we show that

$$\left|V_C(\hat{P}) - W_\eta(\mathbf{p}, \mathbf{q})\right| = \left|V_C(\hat{P}) - V_C(P^\eta)\right| \leqslant \varepsilon. \tag{7}$$

We do so by bounding $|V_C(\hat{P}) - V_C(\tilde{P})| + |V_C(\tilde{P}) - V_C(P^\eta)|$, where $\tilde{P} := D_1 \widetilde{K} D_2$ is the approximate projection computed in Line 3. By Lemma 2, the output of Line 2 satisfies $\|K - \widetilde{K}\|_\infty \leqslant \frac{\varepsilon'}{2} e^{-4\eta R^2}$, and by Lemma L this implies that $\|\log K - \log \widetilde{K}\|_\infty \leqslant \varepsilon'$. Therefore, by Theorem 5, $|V_C(\tilde{P}) - V_C(P^\eta)| \leqslant \frac{\varepsilon}{2}$. Moreover, by Lemma B, $\|\tilde{P} - \hat{P}\|_1 \leqslant \|\tilde{P}\mathbf{1} - \mathbf{p}\|_1 + \|\tilde{P}^T\mathbf{1} - \mathbf{q}\|_1 \leqslant \varepsilon'$, thus by an application of Lemmas E and M, we have that $|V_C(\hat{P}) - V_C(\tilde{P})| \leqslant \frac{\varepsilon}{2}$. Therefore $|V_C(\hat{P}) - V_C(\tilde{P})| + |V_C(\tilde{P}) - V_C(P^\eta)| \leqslant \varepsilon$, which proves (7) and thus also (3a).

Next, we prove (3b). By Proposition 1, $P^\eta = \operatorname{argmin}_{P \in \mathcal{M}(\mathbf{p}, \mathbf{q})} V_C(P)$. Thus

$$\varepsilon \geqslant V_C(\hat{P}) - V_C(P^\eta) = \nabla V_C(P^\eta)(\hat{P} - P^\eta) + \eta^{-1}\mathsf{KL}(\hat{P}\|P^\eta) \geqslant \eta^{-1}\mathsf{KL}(\hat{P}\|P^\eta).$$

where above the first inequality is by (7), the equality is by Lemma F, and the final inequality is by first-order KKT conditions which give $\nabla V_C(P^\eta)(\hat{P} - P^\eta) \geqslant 0$. After rearranging, we conclude that $\mathsf{KL}(\hat{P}\|P^\eta) \leqslant \eta\varepsilon$, proving (3b).

Finally, by Theorem 5, $|\hat{W} - V_C(\tilde{P})| \leqslant \frac{\varepsilon}{2}$, and we have already shown in our proof of (3a) that $|V_C(\tilde{P}) - V_C(P^\eta)| \leqslant \frac{\varepsilon}{2}$, which proves (3c).

**Runtime analysis.** Let $r$ denote the rank of $\tilde{K}$. Note that $r$ is a random variable. By Lemma 2, we have that

$$\mathbb{P}\big(r \leqslant cr^*(X, \eta, \varepsilon') \log \tfrac{n}{\delta}\big) \geqslant 1 - \delta. \tag{8}$$

Now by Lemma 2, the ADAPTIVENYSTRÖM algorithm in line 2 runs in time $O(nr^2)$, and moreover further matrix-vector multiplications with $\tilde{K}$ can be computed in time $O(nr)$. Thus the SINKHORN algorithm in line 3 runs in time $\tilde{O}(nr\eta R^2 \varepsilon'^{-1})$ by Theorem 5, and the ROUND algorithm in line 4 runs in time $O(nr)$ by Lemma B. Combining these bounds and using the choice of $\varepsilon'$ completes the proof of Theorem 1. $\qquad\square$

# 6 Experimental results

In this section we empirically validate our theoretical results. To run our experiments, we used a desktop with 32GB ram and 16 cores Xeon E5-2623 3GHz. The code is optimized in terms of matrix-matrix and matrix-vector products using BLAS-LAPACK primitives.

Fig. 1 plots the time-accuracy tradeoff for NYS-SINK, compared to the standard SINKHORN algorithm. This experiment is run on random point clouds of size $n \approx 20000$, which corresponds to cost matrices of dimension approximately $20000 \times 20000$. Fig. 1 shows that NYS-SINK *is consistently orders of magnitude faster to obtain the same accuracy.*

Next, we investigate NYS-SINK's dependence on the intrinsic dimension and ambient dimension of the input. This is done by running NYS-SINK on distributions supported on 1-dimensional curves embedded in higher dimensions, illustrated in Fig. 2, left. Fig. 2, right, indicates that an approximation rank of $r = 300$ is sufficient to achieve an error

Figure 1: Time-accuracy tradeoff for Nys-Sink and Sinkhorn, for a range of regularization parameters $\eta$ (each corresponding to a different Sinkhorn distance $W_\eta$) and approximation ranks $r$. Each experiment has been repeated 50 times; the variance is indicated by the shaded area around the curves. Note that curves in the plot start at different points corresponding to the time required for initialization.

smaller than $10^{-4}$ for any ambient dimension $5 \leqslant d \leqslant 100$. This empirically validates the result in Corollary 4, namely that *the approximation rank – and consequently the computational complexity of* Nys-Sink *– is independent of the ambient dimension.*

Finally, we evaluate the performance of our algorithm on a benchmark dataset used in computer graphics: we measure Wasserstein distance between 3D cloud points from "The Stanford 3D Scanning Repository"[4]. In the first experiment, we measure the distance between `armadillo` ($n = 1.7 \times 10^5$ points) and `dragon` (at resolution 2, $n = 1.0 \times 10^5$ points), and in the second experiment we measure the distance between `armadillo` and `xyz-dragon` which has more points ($n = 3.6 \times 10^6$ points). The point clouds are centered and normalized in the unit cube. The regularization parameter is set to $\eta = 15$, reflecting the moderate regularization regime typically used in practice.

We compare our algorithm (Nys-Sink)—run with approximation rank $r = 2000$ for $T = 20$ iterations on a GPU—against two algorithms implemented in the library `GeomLoss`[5]. These algorithms are both highly optimized and implemented for GPUs. They are: (a) an algorithm based on an annealing heuristic for $\eta$ (controlled by the parameter $\alpha$, such that at each iteration $\eta_t = \alpha \eta_{t-1}$, see Kosowsky and Yuille, 1994b) and (b) a multiresolution algorithm based on coarse-to-fine clustering of the dataset together with the annealing

Figure 2: Left: one-dimensional curve embedded in $\mathbb{R}^d$, for $d = 3$. For $d \geqslant 4$, the curve we use in dimension $d$ is obtained from the curve we use in the dimension $d-1$ by adding a perpendicular segment of length $1/d^2$ to one endpoint. Right: Accuracy of NYS-SINK as a function of running time, for different ambient dimensions. Each experiment uses a fixed approximation rank $r = 300$.

heuristic (Schmitzer, 2019). Table 1 reports the results, which demonstrate that our method is comparable in terms of precision, and has computational time that is orders of magnitude smaller than the competitors. We note the parameters $r$ and $T$ for NYS-SINK are chosen by hand to balance precision and time complexity.

We note that in these experiments, instead of using Algorithm 2 to choose the rank adaptively, we simply run experiments with a small fixed choice of $r$. As our experiments demonstrate, NYS-SINK achieves good empirical performance even when the rank $r$ is smaller than our theoretical analysis requires. Investigating this empirical success further is an interesting topic for future study.

| Experiment 1: $n \approx 3 \times 10^5$ | $W_\eta$ | time (s) |
|---|---|---|
| Nys-Sink ($r = 2000, T = 20$) | $0.087 \pm 0.008$ | $0.4 \pm 0.1$ |
| Dual-Sink + Annealing ($\alpha = 0.95$) | 0.087 | 35.4 |
| Dual-Sink Multiscale + Annealing ($\alpha = 0.95$) | 0.090 | 3.4 |

| Experiment 2: $n \approx 3.8 \times 10^6$ | $W_\eta$ | time (s) |
|---|---|---|
| Nys-Sink ($r = 2000, T = 20$) | $0.11 \pm 0.01$ | $6.3 \pm 0.8$ |
| Dual-Sink + Annealing ($\alpha = 0.95$) | 0.10 | 1168 |
| Dual-Sink Multiscale + Annealing ($\alpha = 0.95$) | 0.11 | 103.6 |

Table 1: Comparison of our proposed algorithm to existing, highly-optimized GPU-based algorithms, on a large-scale computer graphics benchmark dataset.

# A  Pseudocode for subroutines

## A.1  Pseudocode for Sinkhorn algorithm

As mentioned in the main text, we use the following variant of the classical Sinkhorn algorithm for our theoretical results. Note that in this paper, $\tilde{K}$ is not stored explicitly but instead is stored in factored form. This enables the Sinkhorn algorithm to be implemented quickly since all computations using $\tilde{K}$ are matrix-vector multiplications with $\tilde{K}$ and $\tilde{K}^T$ (see discussion in 3.1 for details).

**Input:** $\tilde{K}$ (in factored form), $\mathbf{p}, \mathbf{q} \in \Delta_n$, $\delta > 0$
**Output:** Positive diagonal matrices $D_1, D_2 \in \mathbb{R}^{n \times n}$, cost $\hat{W}$
 1: $\tau \leftarrow \frac{\delta}{8}$, $D_1, D_2 \leftarrow I_{n \times n}$, $k \leftarrow 0$
 2: $\mathbf{p}' \leftarrow (1 - \tau)\mathbf{p} + \frac{\tau}{n}\mathbf{1}$, $\mathbf{q}' \leftarrow (1 - \tau)\mathbf{q} + \frac{\tau}{n}\mathbf{1}$ &emsp;&emsp;&emsp;&emsp;&emsp; ▷ Round $\mathbf{p}$ and $\mathbf{q}$
 3: **while** $\|D_1\tilde{K}D_2\mathbf{1} - \mathbf{p}'\|_1 + \|(D_1\tilde{K}D_2)^\top\mathbf{1} - \mathbf{q}'\|_1 \leqslant \frac{\delta}{2}$ **do**
 4: &emsp;&emsp; $k \leftarrow k + 1$
 5: &emsp;&emsp; **if** $k$ odd **then**
 6: &emsp;&emsp;&emsp; $(D_1)_{ii} \leftarrow \mathbf{p}'_i/(\tilde{K}D_2\mathbf{1})_i$ for $i = 1, \ldots, n$. &emsp;&emsp;&emsp; ▷ Renormalize rows
 7: &emsp;&emsp; **else**
 8: &emsp;&emsp;&emsp; $(D_2)_{jj} \leftarrow \mathbf{q}'_j/((D_1\tilde{K})^\top\mathbf{1})_j$ for $j = 1, \ldots, n$. &emsp;&emsp; ▷ Renormalize columns
 9: &emsp;&emsp; **end if**
10: **end while**
11: $\hat{W} \leftarrow \sum_{i=1}^n \log(D_1)_{ii}(D_1\tilde{K}D_2\mathbf{1})_i + \sum_{j=1}^n \log(D_2)_{jj}((D_1\tilde{K}D_2)^\top\mathbf{1})_j$
12: **return** $D_1, D_2, \hat{W}$

**Algorithm 3:** SINKHORN

**Lemma A.** *Let $\tilde{C} := -\eta^{-1}\log\tilde{K}$ and let $\tilde{P} := D_1\tilde{K}D_2$, where $D_1$ and $D_2$ are the scaling matrices output by* SINKHORN. *Then the output $\hat{W}$ of* SINKHORN *satisfies $\hat{W} = V_{\tilde{C}}(\tilde{P})$. Moreover, computing $\hat{W}$ takes time $O(\mathrm{T}_{\mathrm{MULT}} + n)$, where $\mathrm{T}_{\mathrm{MULT}}$ is the time required to take matrix-vector products with $\tilde{K}$ and $\tilde{K}^\top$.*

*Proof.* Then

$$\langle\tilde{C}, \tilde{P}\rangle - \eta^{-1}H(\tilde{P}) = \langle\tilde{C}, \tilde{P}\rangle + \eta^{-1}\sum_{i,j=1}^n \tilde{P}_{ij}\log\tilde{P}_{ij}$$

$$= \langle\tilde{C}, \tilde{P}\rangle + \eta^{-1}\sum_{i,j=1}^n \tilde{P}_{ij}(\log(D_1)_{ii} + \log(D_2)_{jj} - \eta\tilde{C}_{ij})$$

$$= \sum_{i,j=1}^n \tilde{P}_{ij}\log(D_1)_{ii} + \sum_{i,j=1}^n \tilde{P}_{ij}\log(D_2)_{jj}$$

$$= \sum_{i=1}^n \log(D_1)_{ii}(\tilde{P}\mathbf{1})_i + \sum_{j=1}^n \log(D_2)_{jj}(\tilde{P}^\top\mathbf{1})_j = \hat{W}.$$

Moreover, the matrices $\log(D_1)$ and $\log(D_2)$ can each be formed in $O(n)$ time, so computing $\hat{W}$ takes time $O(\mathrm{T}_{\mathrm{MULT}} + n)$, as claimed. □

## A.2 &emsp; Pseudocode for rounding algorithm

For completeness, here we briefly recall the rounding algorithm ROUND from (Altschuler et al., 2017) and prove a slight variant of their Lemma 7 that we need for our purposes.

It will be convenient to develop a little notation. For a vector $x \in \mathbb{R}^n$, $\mathbb{D}(x)$ denotes the $n \times n$ diagonal matrix with diagonal entries $[\mathbb{D}(x)]_{ii} = x_i$. For a matrix $A$, $r(A) := A\mathbf{1}$ and $c(A) := A^T\mathbf{1}$ denote the row and column marginals of $A$, respectively. We further denote $r_i(A) = [r(A)]_i$ and similarly $c_j(A) := [c(A)]_j$.

<div style="border:1px solid black; padding:10px;">

**Input:** $F \in \mathbb{R}^{n \times n}$ and $\mathbf{p}, \mathbf{q} \in \Delta_n$
**Output:** $G \in \mathcal{M}(\mathbf{p}, \mathbf{q})$
  1: $X \leftarrow \mathbb{D}(x)$, where $x_i := \frac{\mathbf{p}_i}{r_i(F)} \wedge 1$
  2: $F' \leftarrow XF$
  3: $Y \leftarrow \mathbb{D}(y)$, where $y_j := \frac{\mathbf{q}_j}{c_j(F')} \wedge 1$
  4: $F'' \leftarrow F'Y$
  5: $\text{err}_r \leftarrow \mathbf{p} - r(F'')$, $\text{err}_c \leftarrow \mathbf{q} - c(F'')$
  6: Output $G \leftarrow F'' + \text{err}_r \text{err}_c^T / \|\text{err}_r\|_1$

</div>

**Algorithm 4:** ROUND (from Algorithm 2 in (Altschuler et al., 2017))

**Lemma B.** *If $\mathbf{p}, \mathbf{q} \in \Delta_n$ and $F \in \mathbb{R}_{\geqslant 0}^{n \times n}$, then $\text{ROUND}(F, \mathbf{p}, \mathbf{q})$ outputs a matrix $G \in \mathcal{M}(\mathbf{p}, \mathbf{q})$ of the form $G = D_1 F D_2 + uv^\top$ for positive diagonal matrices $D_1$ and $D_2$ satisfying*

$$\|G - F\|_1 \leqslant \left[ \|F\mathbf{1} - \mathbf{p}\|_1 + \|F^T\mathbf{1} - \mathbf{q}\|_1 \right].$$

*Moreover, the algorithm only uses $O(1)$ matrix-vector products with $F$ and $O(n)$ additional processing time.*

*Proof.* The runtime claim is clear. Next, let $\Delta := \|F\|_1 - \|F''\|_1 = \sum_{i=1}^n (r_i(F) - \mathbf{p}_i)_+ + \sum_{j=1}^n (c_j(F') - \mathbf{q}_j)_+$ denote the amount of mass removed from $F$ to create $F''$. Observe that $\sum_{i=1}^n (r_i(F) - \mathbf{p}_i)_+ = \frac{1}{2} \|r(F) - \mathbf{p}\|_1$. Since $F' \leqslant F$ entrywise, we also have $\sum_{j=1}^n (c_j(F') - \mathbf{q}_j)_+ \leqslant \sum_{j=1}^n (c_j(F) - \mathbf{q}_j)_+ = \frac{1}{2} \|c(F) - \mathbf{q}\|_1$. Thus $\Delta \leqslant \frac{1}{2}(\|r(F) - \mathbf{p}\|_1 + \|c(F) - q\|_1)$. The proof is complete since $\|F - G\|_1 \leqslant \|F - F''\|_1 + \|F'' - G\|_1 = 2\Delta$. $\qquad\square$

# B  Omitted proofs

## B.1  Stability inequalities for Sinkhorn distances

**Lemma C.** *Let $C, \tilde{C} \in \mathbb{R}^{n \times n}$. If $P \in \Delta_{n \times n}$, then*

$$|V_C(P) - V_{\tilde{C}}(P)| \leqslant \|C - \tilde{C}\|_\infty.$$

*Proof.* By Hölder's inequality, $|V_C(P) - V_{\tilde{C}}(P)| = |\langle C - \tilde{C}, P \rangle| \leqslant \|C - \tilde{C}\|_\infty \|P\|_1 = \|C - \tilde{C}\|_\infty$. $\qquad\square$

**Lemma D.** *Let $P, Q \in \Delta_{n \times n}$. If $\|P - Q\|_1 \leqslant \delta \leqslant 1$, then*

$$|H(P) - H(Q)| \leqslant \delta \log \frac{2n}{\delta}.$$

*Proof.* By Ho and Yeung (2010, Theorem 6), $|H(P) - H(Q)| \leqslant \frac{\delta}{2} \log(n^2 - 1) + h\left(\frac{\delta}{2}\right)$, where $h$ is the binary entropy function. If $\delta \leqslant 1$, then $h(\frac{\delta}{2}) \leqslant \delta \log \frac{2}{\delta}$, which yields the claim. $\qquad\square$

**Lemma E.** *Let $M \in \mathbb{R}^{n \times n}$, $\eta > 0$, and $P, Q \in \Delta_{n \times n}$. If $\|P - Q\|_1 \leqslant \delta \leqslant 1$, then*

$$|V_M(P) - V_M(Q)| \leqslant \delta \|M\|_\infty + \eta^{-1} \delta \log \frac{2n}{\delta}.$$

*Proof.* By definition of $V_M(\cdot)$ and the triangle inequality, $|V_M(P) - V_M(Q)| \leqslant |\sum_{ij}(P_{ij} - Q_{ij})M_{ij}| + \eta^{-1}|H(P) - H(\tilde{P})|$. By Hölder's inequality, the former term is upper bounded by $\|P - Q\|_1 \|M\|_\infty \leqslant \delta \|M\|_\infty$. By Lemma D, the latter term above is upper bounded by $\eta^{-1}\delta \log \frac{2n}{\delta}$. $\qquad\square$

## B.2 Bregman divergence of Sinkhorn distances

The remainder in the first-order Taylor expansion of $V_C(\cdot)$ between any two joint distributions is exactly the KL-divergence between them.

**Lemma F.** *For any $C \in \mathbb{R}^{n \times n}$, $\eta > 0$, and $P, Q \in \Delta^{n \times n}$,*

$$V_C(Q) = V_C(P) + \langle \nabla V_C(P), (Q - P) \rangle + \eta^{-1}\mathsf{KL}(Q \| P).$$

*Proof.* Observing that $\nabla V_C(P)$ has $ij$th entry $C_{ij} + \eta^{-1}(1 + \log P_{ij})$, we expand the right hand side as $[\langle C, P \rangle + \eta^{-1}\sum_{ij} P_{ij} \log P_{ij}] + [\langle C, Q - P \rangle + \eta^{-1}\sum_{ij}(Q_{ij} - P_{ij})\log P_{ij}] + [\eta^{-1}\sum_{ij} Q_{ij} \log \frac{Q_{ij}}{P_{ij}}] = \langle C, Q \rangle + \eta^{-1}\sum_{ij} Q_{ij} \log Q_{ij} = V_C(Q).$ $\qquad\square$

## B.3 Hausdorff distance between transport polytopes

**Lemma G.** *Let $d_H$ denote the Hausdorff distance with respect to $\|\cdot\|_1$. If $\mathbf{p}, \tilde{\mathbf{p}}, \mathbf{q}, \tilde{\mathbf{q}} \in \Delta_n$, then*

$$d_H(\mathcal{M}(\mathbf{p}, \mathbf{q}), \mathcal{M}(\tilde{\mathbf{p}}, \tilde{\mathbf{q}})) \leqslant \|\mathbf{p} - \tilde{\mathbf{p}}\|_1 + \|\mathbf{q} - \tilde{\mathbf{q}}\|_1 .$$

*Proof.* Follows immediately from Lemma B. $\qquad\square$

**Lemma H.** *Fix a norm $\|\cdot\|$ on $\mathcal{X}$. If $f : \mathcal{X} \to \mathbb{R}$ satisfies $|f(x) - f(y)| \leqslant \omega(\|x - y\|)$ for $\omega$ an increasing, upper semicontinuous function, then for any two sets $A, B \subseteq \mathcal{X}$,*

$$\left| \inf_{x \in A} f(x) - \inf_{x \in B} f(x) \right| \leqslant \omega(d_H(A, B)),$$

*where $d_H(A, B)$ is the Hausdorff distance between $A$ and $B$ with respect to $\|\cdot\|$.*

*Proof.*

$$\inf_{x \in A} f(x) - \inf_{x \in B} f(x) \leqslant \sup_{y \in B} \inf_{x \in A} f(x) - f(y)$$
$$\leqslant \sup_{y \in B} \inf_{x \in A} \omega(\|x - y\|)$$
$$\leqslant \omega(\sup_{y \in B} \inf_{x \in A} \|x - y\|)$$
$$\leqslant \omega(d_H(A, B)).$$

Interchanging the role of $A$ and $B$ yields the claim. $\qquad\square$

## B.4    Miscellaneous helpful lemmas

**Lemma I.** *Let $X \subset \mathbb{R}^d$ be convex, and let $f : X \to \mathbb{R}$ be 1-strongly-convex with respect to some norm $\|\cdot\|$. If $x_a^* = \operatorname{argmin}_{x \in \mathcal{X}} \langle a, x \rangle + f(x)$ and $x_b^* = \operatorname{argmin}_{x \in \mathcal{X}} \langle b, x \rangle + f(x)$, then*

$$\|x_a^* - x_b^*\| \leqslant \|a - b\|_* \,,$$

*where $\|\cdot\|_*$ denotes the dual norm to $\|\cdot\|$.*

*Proof.* This amounts to the well known fact (see, e.g., Hiriart-Urruty and Lemaréchal, 2001, Theorem 4.2.1) that the Legendre transform of a strongly convex function has Lipschitz gradients. We assume without loss of generality that $f = +\infty$ outside of $\mathcal{X}$, so that $f$ can be extended to a function on all of $\mathbb{R}^d$ and thus we can take the minima to be unconstrained. The fact that $f(y) + \langle y, a \rangle \geqslant f(x_a^*) + \langle x_a^*, a \rangle$ for all $y$ implies that $-a \in \partial f(x_a^*)$, and likewise $-b \in \partial f(x_b^*)$. Thus by definition of strong convexity, we have

$$f(x_a^*) \geqslant f(x_b^*) + \langle -b, x_a^* - x_b^* \rangle + \frac{1}{2} \|x_a^* - x_b^*\|^2,$$

$$f(x_b^*) \geqslant f(x_a^*) + \langle -a, x_b^* - x_a^* \rangle + \frac{1}{2} \|x_b^* - x_a^*\|^2,$$

Adding these inequalities yields

$$\langle b - a, x_a^* - x_b^* \rangle \geqslant \|x_a^* - x_b^*\|^2 \,,$$

which implies the claim via the definition of the dual norm. $\qquad\square$

**Lemma J.** *For any matrix $A \in \mathbb{R}^{n \times n}$,*

$$\|A\|_\infty \leqslant \|A\|_{\mathrm{op}} \leqslant n\|A\|_\infty$$

*Proof.* By duality between the operator norm and the nuclear norm, $\|A\|_\infty = \max_{i,j \in [n]} |e_i^T A e_j| \leqslant \max_{i,j \in [n]} \|A\|_{\mathrm{op}} \|e_i e_j^T\|_* = \|A\|_{\mathrm{op}}$. This establishes the first inequality.

Next, for any $v \in \mathbb{R}^n$ with unit norm $\|v\|_2 = 1$, note that $\|Av\|_2^2 = \sum_{i=1}^n \left( \sum_{j=1}^n A_{ij} v_j \right)^2 \leqslant \sum_{i=1}^n n\|A\|_\infty^2 \sum_{j=1}^n v_j^2 = n^2 \|A\|_\infty^2$, proving the second inequality. $\qquad\square$

**Lemma K.** *For any $a, b > 0$,*

$$|\log a - \log b| \leqslant \frac{|a - b|}{\min\{a, b\}} \,.$$

*Proof.* Without loss of generality, assume $a \geqslant b$. Then $\log a - \log b = \log \frac{a}{b} \leqslant \frac{a}{b} - 1 = \frac{a-b}{\min\{a,b\}}$, as claimed. $\qquad\square$

**Lemma L.** *Let $\{x_1, \ldots, x_n\} \subset \mathbb{R}^d$ lie in an Euclidean ball of radius $R$, and let $\eta > 0$. Denote by $K \in \mathbb{R}^{n \times n}$ the matrix with entries $K_{ij} := e^{-\eta \|x_i - x_j\|_2^2}$. If a matrix $\widetilde{K} \in \mathbb{R}^{n \times n}$ satisfies $\|K - \widetilde{K}\|_\infty \leqslant \frac{\varepsilon'}{2} e^{-4\eta R^2}$ for some $\varepsilon' \in (0, 1)$, then*

$$\|\log K - \log \widetilde{K}\|_\infty \leqslant \varepsilon' \,.$$

*Proof.* Since $\|x_i\|_2 \leqslant R$ for all $i \in [n]$, the matrix $K$ satisfies $K_{ij} = e^{-\eta\|x_i-x_j\|_2^2} \geqslant e^{-4\eta R^2}$ for all $i, j \in [n]$. Hence $\widetilde{K}_{ij} \geqslant \frac{\varepsilon'}{2} e^{-4\eta R^2}$ for all $i, j \in [n]$ and thus by Lemma K,

$$|\log K_{ij} - \log \widetilde{K}_{ij}| \leqslant \frac{|K_{ij} - \widetilde{K}_{ij}|}{\min\{\widetilde{K}_{ij}, K_{ij}\}} \leqslant \varepsilon'.$$

$\square$

**Lemma M.** *Let* $n \in \mathbb{N}$, $\varepsilon \in (0,1)$, $\|C\|_\infty \geqslant 1$, *and* $\eta \in [1,n]$. *Then for any* $\delta \leqslant \frac{\eta\varepsilon}{50(\|C\|_\infty\eta+\log\frac{n}{\eta\varepsilon})}$, *the bound* $\delta(2\|C\|_\infty + 3\eta^{-1}) + 2\eta^{-1}\delta\log\frac{2n}{\delta} \leqslant \frac{\varepsilon}{2}$ *holds.*

*Proof.* We write

$$\delta(2\|C\|_\infty + 3\eta^{-1}) + 2\eta^{-1}\delta\log\frac{2n}{\delta} = \delta(2\|C\|_\infty + 3\eta^{-1}) + 2\eta^{-1}\delta\log\frac{2n}{\eta\varepsilon} + 2\eta^{-1}\delta\log\frac{\eta\varepsilon}{\delta}$$

and bound the three terms separately. First, the assumptions imply that $\eta\varepsilon \leqslant n$ and $2\|C\|_\infty + 3\eta^{-1} \leqslant 5\|C\|_\infty$. We therefore have

$$\delta(2\|C\|_\infty + 3\eta^{-1}) \leqslant \frac{5\|C\|_\infty\eta\varepsilon}{50(\|C\|_\infty\eta + \log\frac{n}{\eta\varepsilon})} \leqslant \frac{1}{10}\varepsilon.$$

Since $\|C\|_\infty\eta \geqslant 1$, we likewise obtain

$$2\eta^{-1}\delta\log\frac{2n}{\eta\varepsilon} \leqslant \frac{2\varepsilon\log\frac{2n}{\eta\varepsilon}}{50(1+\log\frac{n}{\eta\varepsilon})} = \frac{2(\log 2 + \log\frac{n}{\eta\varepsilon})}{50(1+\log\frac{n}{\eta\varepsilon})}\varepsilon \leqslant \frac{1}{25}\varepsilon.$$

Finally, the fact that $\frac{\eta^{-1}\delta}{\varepsilon} \leqslant \frac{1}{50}$ and $x\log\frac{1}{x} \leqslant \frac{1}{10}$ for $x \leqslant \frac{1}{50}$ yields

$$2\eta^{-1}\delta\log\frac{\eta\varepsilon}{\delta} = 2\left(\frac{\eta^{-1}\delta}{\varepsilon}\log\frac{\varepsilon}{\eta^{-1}\delta}\right)\varepsilon \leqslant \frac{1}{5}\varepsilon.$$

$\square$

## B.5 Supplemental results for Section 3

### B.5.1 Full proof of Theorem 3

Define $\phi_\alpha(x) := (2\eta)^{\sum_{j=1}^d \alpha_j/2}\prod_{j=1}^d[(\alpha_j!)^{-1/2}x_j^{\alpha_j}e^{-\eta x_j^2}]$, for $x \in \mathbb{R}^d$ and $\alpha \in (\mathbb{N} \cup \{0\})^d$, and define $\psi_T(x) := (\phi_\alpha(x))_{\alpha_1+\cdots+\alpha_d\leqslant T}$. Note that $\psi_T : \mathbb{R}^d \to \mathbb{R}^M$, with $M = \binom{d+T}{T}$. By Cotter et al. (2011, equation 11), we have

$$\sup_{x,x'\in B_R^d}|k_\eta(x,x') - \psi_T(x)^\top\psi_T(x')| \leqslant \frac{(2\eta R^2)^{T+1}}{(T+1)!} =: \varepsilon(T).$$

Now denote by $\Psi_T \in \mathbb{R}^{M\times n}$ the matrix $\Psi_T := (\psi_T(x_1), \ldots, \psi_T(x_n))$. By Lemma J, we have

$$\|K - \Psi_T^\top\Psi_T\|_{\mathrm{op}} \leqslant n\sup_{i,j}|k_\eta(x_i,x_j) - \psi_T(x_i)^\top\psi_T(x_j)| \leqslant n\varepsilon(T).$$

By the Eckart-Young-Mirsky Theorem, we have

$$\lambda_{M+1} = \inf_{\bar{K}_T\in\mathbb{R}^{n\times n},\,\mathrm{rank}(\bar{K}_T)\leqslant M}\|K - \bar{K}_T\|_{\mathrm{op}}.$$

Therefore by combining the above two displays, we conclude that

$$\lambda_{M+1} \leqslant \|K - \Psi_T^\top\Psi_T\|_{\mathrm{op}} \leqslant n\varepsilon(T).$$

**Point 1.** We recall that for any $d, q \in \mathbb{N}$, the inequality $\binom{d+q}{q} \leqslant e^d(1 + q/d)^d$ holds. Therefore, given $t \geqslant (2e)^d$, choosing $T = \lfloor dt^{1/d}/(2e) \rfloor$ yields $\binom{T+d}{d} \leqslant e^d(1 + T/d)^d \leqslant t$. We therefore have $\lambda_{t+1} \leqslant n\varepsilon(T)$ for this choice of $T$. Now, by Stirling's approximation of $(T+1)!$, we have that $\varepsilon(T) \leqslant e^{-(T+1)\log\frac{T+1}{2e\eta R^2}}$. If $T \geqslant 2e\eta R^2$, then $(T+1)\log\frac{T+1}{2e\eta R^2} \geqslant \frac{dt^{1/d}}{2e}\log\frac{dt^{1/d}}{4e^2\eta R^2}$, which yields the desired bound. On the other hand, when $T < 2e\eta R^2$, we use the trivial bound $\lambda_t(K) \leqslant \mathrm{Tr}(K) \leqslant n$. The claim follows.

**Point 2.** We have that $\lambda_{M_T+1}(K) \leqslant n\varepsilon(T)$, for $M_T = \binom{d+T}{T}$ and $T \in \mathbb{N}$. Since the eigenvalues are in decreasing order we have that $\lambda_{M_{T+1}+1} \leqslant \lambda_t(K) \leqslant \lambda_{M_T+1}(K)$ for $M_T + 1 \leqslant t \leqslant M_{T+1} + 1$. Since $x/(x+\tau)$ is increasing in $x$, for $x \geqslant 0$, we have

$$\sum_{t=1}^n \frac{\lambda_j(K)}{\lambda_j(K) + n\tau} \leqslant \sum_{T=0}^\infty (M_{T+1} - M_T)\frac{\lambda_{M_T+1}(K)}{\lambda_{M_T+1}(K) + n\tau} \leqslant \sum_{T=0}^\infty (M_{T+1} - M_T)\frac{\varepsilon(T)}{\varepsilon(T) + \tau}.$$

Let $T_\tau$ be such that $\varepsilon(T_\tau) \leqslant \tau$. We can then bound $\varepsilon(T)/(\varepsilon(T) + \tau)$ above by 1 for $T \leqslant T_\tau - 1$ and by $\varepsilon(T)/\tau$ for $T \geqslant T_\tau$, obtaining

$$\sum_{T=0}^\infty (M_{T+1} - M_T)\frac{\varepsilon(T)}{\varepsilon(T) + \tau} \leqslant \sum_{T=0}^{T_\tau - 1}(M_{T+1} - M_T) + \sum_{T=T_\tau}^\infty (M_{T+1} - M_T)\frac{\varepsilon(T)}{\tau}$$

$$= M_{T_\tau} + \frac{1}{\tau}\sum_{T=T_\tau}^\infty (M_{T+1} - M_T)\varepsilon(T).$$

In particular, we can choose $T_\tau = d + 2e^2\eta R^2 + \log(1/\tau)$. Since $\log\frac{T_\tau}{2e\eta R^2} > 1$, for any $T \geqslant T_\tau$, then $\varepsilon(T) \leqslant e^{-T\log\frac{T}{2e\eta R^2}} \leqslant e^{-T}$. Moreover since $M_{T+1} - M_T = dM_T/(T+1)$, and $M_T \leqslant e^d(1 + T/d)^d$, we have

$$\sum_{T=T_\tau}^\infty (M_{T+1} - M_T)\varepsilon(T) \leqslant \frac{d}{T_\tau}\sum_{T=T_\tau}^\infty M_T e^{-T} \leqslant \frac{de^d}{T_\tau}\sum_{T=T_\tau}^\infty \left(1 + \frac{T}{d}\right)^d e^{-T}$$

$$\leqslant \frac{de^d}{T_\tau}\int_{T_\tau}^\infty \left(1 + \frac{x}{d}\right)^d e^{-x}dx.$$

Finally, by changing variables, $x = u + T_\tau$ and $u = (d + T_\tau)z$,

$$\int_{T_\tau}^\infty \left(1 + \frac{x}{d}\right)^d e^{-x}dx = \int_0^\infty \left(1 + \frac{T_\tau}{d} + \frac{u}{d}\right)^d e^{-u-T_\tau}du$$

$$= \left(1 + \frac{T_\tau}{d}\right)^d e^{-T_\tau}\int_0^\infty \left(1 + \frac{u}{d + T_\tau}\right)^d e^{-u}du$$

$$= \left(1 + \frac{T_\tau}{d}\right)^d e^{-T_\tau}(d + T_\tau)\int_0^\infty (1 + z)^d e^{-(d+T_\tau)z}dz$$

$$= d^{-d}e^d\Gamma(d+1, d+T_\tau),$$

where for the last equality we used the characterization of the incomplete gamma function $\Gamma(a, z) = z^{-a}e^{-z}\int_0^\infty (1+t)^{a-1}e^{-zt}dt$ (see Eq. 8.6.5 of Olver et al., 2010). To complete

the proof note that by Lemma P we have $\Gamma(a, z) \leqslant z/(z-a)z^{a-1}e^{-z}$, for any $z > a > 0$. Since $\log(1/\tau) \geqslant 0$ for $\tau \in (0, 1]$, we have $(d + T_\tau)/(T_\tau - 1) \leqslant 2$ and $(de^{-T_\tau})/(\tau T_\tau) \leqslant 1$, so

$$
\begin{aligned}
d_{\mathrm{eff}}(\tau) &\leqslant M_{T_\tau} + \frac{de^d}{\tau T_\tau} d^{-d} e^d \Gamma(d+1, d+T_\tau) \\
&\leqslant e^d (1 + T_\tau/d)^d \left( 1 + \frac{de^{-T_\tau}}{\tau T_\tau} \frac{d + T_\tau}{T_\tau - 1} \right) \\
&\leqslant 3e^d \left( 2 + \frac{2e^2}{d} \eta R^2 + \frac{1}{d} \log \frac{1}{\tau} \right)^d.
\end{aligned}
$$

$\square$

### B.5.2 Full proof of Theorem 4

The proof of Points 2 and 3 here is completely analogous to the proof of Points 1 and 2, respectively, in Theorem 3.

**Point 2.** Let $\bar{X} = \{\bar{x}_1, \ldots, \bar{x}_p\} \subset \Omega$ be a minimal $\varepsilon$ net of $\Omega$. Since $\Omega$ is a smooth manifold of dimension $k$, then there exists $C_0 > 0$ for which $p \leqslant C_0 \varepsilon^{-k}$. Now let $\bar{K} \in \mathbb{R}^{p \times p}$ be given by $\bar{K}_{i,j} = k_\eta(\bar{x}_i, \bar{x}_j)$ and define $\Phi(x) = \bar{K}^{-1/2} v(x)$, with $v(x) = (k_\eta(x, \bar{x}_1), \ldots, k_\eta(x, \bar{x}_p))$. Then when $f(x') := k_\eta(x', x)$, then $\widehat{f}_{\bar{X}} = \Phi(x')^\top \Phi(x)$. By applying Point 1 to $f(x) = k_\eta(x', x)$, we have

$$
|k_\eta(x', x) - \Phi(x')\Phi(x)| \leqslant e^{-c\varepsilon^{-2/5}}, \qquad \forall x, x' \in \Omega.
$$

If we let $B \in \mathbb{R}^{p \times n} = (\Phi(x_1), \ldots \Phi(x_n))$, then

$$
\|K - B^\top B\|_{\mathrm{op}} \leqslant n \max_{ij} |k_\eta(x', x) - \Phi(x')\Phi(x)| \leqslant n e^{-c\varepsilon^{-2/5}}.
$$

Since $B$ is of rank $p$, the the Eckart-Young-Mirsky Theorem again implies $\lambda_{p+1}(K) \leqslant n e^{-c\varepsilon^{-2/5}}$. We conclude by recalling that $\varepsilon \leqslant (p/C_0)^{-1/k}$.

**Point 3.** Let $M_\tau$ be such that $\lambda_{M_\tau+1} \leqslant n\tau$. By Point 2, this holds if we take $M_\tau = (c_0 \log \frac{1}{\tau})^{5k/2}$ for a sufficiently large constant $c$. By definition of $d_{\mathrm{eff}}(\tau)$ and the fact that $x/(x+\lambda) \leqslant \min(1, x/\lambda)$ for any $x \geqslant 0, \lambda > 0$, we have

$$
\begin{aligned}
d_{\mathrm{eff}}(\tau) &= \sum_{j=1}^n \frac{\lambda_j(K)}{\lambda_j(K) + n\tau} = \sum_{j=1}^{M_\tau+1} \frac{\lambda_j(K)}{\lambda_j(K) + n\tau} + \sum_{j=M_\tau+2}^n \frac{\lambda_j(K)}{\lambda_j(K) + n\tau} \\
&\leqslant M_\tau + 1 + \frac{1}{\tau} \sum_{j=M_\tau+2}^n \frac{\lambda_j(K)}{n} \leqslant (c_0 \log \frac{1}{\tau})^{5k/2} + 1 + \frac{1}{\tau} \sum_{j=M_\tau+1}^\infty e^{-cj^{\frac{2}{5k}}}
\end{aligned}
$$

Denoting $\beta := \frac{2}{5k}$ for shorthand, we can upper bound the sum as follows:

$$\sum_{j=M_\tau+1}^{\infty} e^{-cj^{\frac{2}{5k}}} \leqslant \int_{M_\tau}^{\infty} e^{-cx^\beta} dx$$

$$= \frac{1}{\beta c^{1/\beta}} \int_0^{\infty} u^{\frac{1-\beta}{\beta}} \mathbb{1}(u \geqslant cM_\tau^\beta) e^{-u} du$$

$$\leqslant \frac{1}{\beta c^{1/\beta}} \left( \int_0^{\infty} u^{2(\frac{1-\beta}{\beta})} e^{-u} du \right)^{1/2} \left( \int_{cM_\tau^\beta}^{\infty} e^{-u} du \right)^{1/2}$$

$$= c_k e^{-\frac{1}{2} cM_\tau^\beta} \leqslant c_k \tau,$$

where above the second step was by the change of variables $u := cx^\beta$, the third step was by Cauchy-Schwartz with respect to the inner product $\langle f, g \rangle := \int_0^{\infty} f(u)g(u)e^{-u} du$, and the final line was for some constant $c_k$ only depending on $k$, whenever $c_0$ is taken to be at least $\frac{2}{c}$. This proves the claim. $\qquad\square$

### B.5.3 Additional bounds

**Lemma N.** *Let $A : B \to U$, be a smooth map, with $B \subseteq \mathbb{R}^d$, $U \subseteq \mathbb{R}^m$, $d, m \in \mathbb{N}$, such that there exists $Q > 0$, for which*

$$\|D^\alpha A\|_{L^\infty(B)} \leqslant Q^{|\alpha|}, \quad \forall \alpha \in \mathbb{N}^d,$$

*then, for $\nu \in \mathbb{N}_0^d$, $p \geqslant 1$,*

$$\|D^\nu(f \circ A)\|_{L^p(B)} \leqslant (2|\nu|mQ)^{|\nu|} \max_{|\lambda| \leqslant |\nu|} \|(D^\lambda f) \circ A\|_{L^p(B)}.$$

*Proof.* First we study $D^\nu(f \circ A)$. Let $n := |\nu|$ and $A = (a_1, \ldots, a_m)$ with $a_j : \mathbb{R}^d \to \mathbb{R}$. By the *multivariate Faà di Bruno formula* (Constantine and Savits, 1996), we have that

$$D^\nu(f \circ A) = \nu! \sum_{1 \leqslant |\lambda| \leqslant n} (D^\lambda f) \circ A \sum_{\binom{k_1, \ldots, k_n}{l_1, \ldots, l_n} \in p(\lambda, \nu)} \prod_{j=1}^{n} \prod_{i=1}^{m} \frac{[D^{l_j} a_i]^{[k_j]_i}}{[k_j]_i! l_j!},$$

where the set $p(\lambda, \nu)$ is defined in Constantine and Savits (1996, Eq. 2.4), with $l_1, \ldots, l_n \in \mathbb{N}_0^d$, $k_1, \ldots, k_n \in \mathbb{N}_0^m$ and satisfying $\sum_{j=1}^{n} |k_j| l_j = \nu$. Now by assumption $\|D^{l_j} a_i\| \leqslant Q^{|l_j|}$ for $1 \leqslant i \leqslant m$. Then

$$\left\| \prod_{j=1}^{n} \prod_{i=1}^{m} \frac{[D^{l_j} a_i]^{[k_j]_i}}{[k_j]_i! l_j!} \right\|_{L^\infty(B)} \leqslant Q^{\sum_{j=1}^{n} |l_j||k_j|} \prod_{j=1}^{n} \frac{1}{k_j! l_j!^{|k_j|}}.$$

Now note that by the properties of $l_j, k_j$, we have that $|\nu| = |\sum_j |k_j| l_j| = \sum_j |k_j||l_j|$, then

$$\|D^\nu(f \circ A)\|_{L^p(B)} \leqslant Q^{|\nu|} \max_{|\lambda| \leqslant |\nu|} \|(D^\lambda f) \circ A\|_{L^p(B)} \quad \times \quad \nu! \sum_{1 \leqslant |\lambda| \leqslant n} \sum_{\substack{k_1, \ldots, k_n \\ l_1, \ldots, l_n \in p(\lambda, \nu)}} \prod_{j=1}^{n} \frac{1}{k_j! l_j!^{|k_j|}}.$$

To conclude, denote by $S_k^n$ the *Stirling numbers of the second kind.* By Constantine and Savits (1996, Corollary 2.9) and Rennie and Dobson (1969) we have

$$\nu! \sum_{1 \leqslant |\lambda| \leqslant n} \sum_{\substack{k_1, \ldots, k_n \\ l_1, \ldots, l_n \in p(\lambda, \nu)}} \prod_{j=1}^{n} \frac{1}{k_j! l_j!^{|k_j|}} = \sum_{i=1}^{n} m^k S_k^n \leqslant m^n \sum_{i=1}^{n} \binom{n}{k} k^{n-k} \leqslant m^n (2n)^n.$$

$\square$

**Lemma O.** *Let $\Psi_j : U_j \to B_{r_j}^k$ such that there exists $Q > 0$ for which $\|D^\alpha \Psi_j^{-1}\|_{L^\infty(B_{r_j}^k)} \leqslant Q^{|\alpha|}$ for $\alpha \in \mathbb{N}^k$. Then for any $q \geqslant k$, we have*

$$\|f \circ \Psi_j^{-1}\|_{W_2^q(B_{r_j}^k)} \leqslant C_{d,k,R,r_j} q^k (2qdQ)^q \|f\|_{W_2^{q+(d+1)/2}(B_R^d)}.$$

*Proof.* First note that $\| \cdot \|_{L^\infty(B_R^d)} \leqslant C_{d,R} \| \cdot \|_{W_2^{(d+1)/2}(B_R^d)}$ (Adams and Fournier, 2003) for a constant $C_{d,R}$ depending only on $d$ and $R$. Therefore

$$\begin{aligned}
\|(D^\alpha f) \circ \Psi_j^{-1}\|_{L^2(B_{r_j}^k)} &\leqslant \text{vol}(B_{r_j}^k)^{1/2} \|(D^\alpha f) \circ \Psi_j^{-1}\|_{L^\infty(B_{r_j}^k)} \\
&= \text{vol}(B_{r_j}^k)^{1/2} \|D^\alpha f\|_{L^\infty(U_j)} \leqslant \text{vol}(B_{r_j}^k)^{1/2} \|D^\alpha f\|_{L^\infty(B_R^d)} \\
&\leqslant C_{d,R} \text{vol}(B_{r_j}^k)^{1/2} \|D^\alpha f\|_{W_2^{(d+1)/2}(B_R^d)}.
\end{aligned}$$

Moreover note that $\|D^\alpha f\|_{W_2^{(d+1)/2}(B_R^d)} \leqslant \|f\|_{W_2^{|\alpha|+(d+1)/2}(B_R^d)}$. By Lemma N we have that

$$\|D^\alpha(f \circ \Psi_j^{-1})\|_{L^2(B_{r_j}^k)} \leqslant (2|\alpha|dQ)^{|\alpha|} \max_{|\lambda| \leqslant |\alpha|} \|(D^\lambda f) \circ \Psi_j^{-1}\|_{L^2(B_{r_j}^k)}.$$

By definition of Sobolev space $W_2^q(B_{r_j}^k)$, we have

$$\begin{aligned}
\|f \circ \Psi_j^{-1}\|_{W_2^q(B_{r_j}^k)} &\leqslant \sum_{|\alpha| \leqslant q} \|D^\alpha(f \circ \Psi_j^{-1})\|_{L^2(B_{r_j}^k)} \\
&\leqslant \sum_{|\alpha| \leqslant q} (2|\alpha|dQ)^{|\alpha|} \max_{|\lambda| \leqslant |\alpha|} \|(D^\lambda f) \circ \Psi_j^{-1}\|_{L^2(B_{r_j}^k)} \\
&\leqslant C_q \max_{|\lambda| \leqslant q} \|(D^\lambda f) \circ \Psi_j^{-1}\|_{L^2(B_{r_j}^k)}^2,
\end{aligned}$$

where $C_q := \sum_{|\alpha| \leqslant q} (2|\alpha|dQ)^{|\alpha|}$. Then,

$$\|f \circ \Psi_j^{-1}\|_{W_2^q(B_{r_j}^k)}^2 \leqslant C_q C_{d,R} \text{vol}(B_{r_j}^k)^{1/2} \|f\|_{W_2^{q+(d+1)/2}(B_R^d)}.$$

The final result is obtained via the bound $C_q \leqslant (2qdQ)^q \binom{k+q}{k} \leqslant (2ek)^k (2qdQ)^q q^k$ for $q \geqslant k$. $\square$

**Lemma P** (Bounds for the incomplete gamma function). *Denote by $\Gamma(a, x)$ the function defined as*

$$\Gamma(a, x) = \int_x^\infty z^{a-1} e^{-z} dz,$$

*for $a \in \mathbb{R}$ and $x > 0$. When $x \geqslant (a-1)_+ \ \wedge \ x > 0$, the following holds*

$$\Gamma(a,x) \leqslant \frac{x}{x - (a-1)_+} x^{a-1} e^{-x},$$

*In particular $\Gamma(a,x) \leqslant 2x^{a-1}e^{-x}$, for $x \geqslant 2(a-1)_+ \ \wedge \ x > 0$.*

*Proof.* Assume $x > 0$. When $a \leqslant 1$, the function $z^{a-1}e^{-z}$ is decreasing and in particular $z^{a-1}e^{-z} \leqslant x^{a-1}e^{-z}$ for $z \geqslant x$, so when $z \geqslant x$ we have

$$\Gamma(a,x) = \int_x^\infty z^{a-1}e^{-z}dz \leqslant x^{a-1}\int_x^\infty e^{-z}dz = x^{a-1}e^{-x}.$$

When $a > 1$, for any $\tau \in (0,1)$, we have

$$\Gamma(a,x) = \int_x^\infty z^{a-1}e^{-z}dz = \int_x^\infty (z^{a-1}e^{-\tau z})(e^{-(1-\tau)z})dz$$

$$\leqslant \sup_{z \geqslant x}(z^{a-1}e^{-\tau z}) \int_x^\infty e^{-(1-\tau)z}dz = \frac{e^{-(1-\tau)x}}{1-\tau} \sup_{z \geqslant x}(z^{a-1}e^{-\tau z}).$$

Now note that the maximum of $z^{a-1}e^{-\tau z}$ is reached when $z = (a-1)/\tau$. When $x \geqslant a-1$, we can set $\tau = (a-1)/x$, so the maximum of $z^{a-1}e^{-\tau z}$ is exactly in $z = x$. In that case $\sup_{z \geqslant x}(z^{a-1}e^{-\tau z}) = x^{a-1}e^{-\tau x}$ and

$$\Gamma(a,x) \leqslant \frac{x}{x - (a-1)} x^{a-1} e^{-x}.$$

The final result is obtained by gathering the cases $a \leqslant 1$ and $a > 0$ in the same expression. $\qquad \square$

**Corollary A.** *Let $a \in \mathbb{R}$, $A, q_1, q_2, b > 0$. When $q_2 A^b \geqslant 2\left(\frac{a+1}{b} - 1\right)_+$, the following holds*

$$\int_A^\infty q_1 x^a e^{-q_2 x^b} dx \leqslant \frac{2q_1}{bq_2} A^{a+1-1/b} e^{-q_2 A^b}.$$

*Proof.* By the change of variable $x = (u/q_2)^{1/b}$ we have

$$\int_A^\infty q_1 x^a e^{-q_2 x^b} dx = \frac{q_1 q_2^{-\frac{a+1}{b}}}{b} \int_{q_2 A^b}^\infty u^{\frac{a+1}{b}-1} e^{-u} du$$

$$= \frac{q_1 q_2^{-\frac{a+1}{b}}}{b} \Gamma\left(\frac{a+1}{b}, q_2 A^b\right)$$

$$\leqslant \frac{2q_1}{bq_2} A^{a+1-1/b} e^{-q_2 A^b},$$

where for the last step we used Lemma P. $\qquad \square$

# C  Lipschitz properties of the Sinkhorn projection

We give a simple construction illustrating that the Sinkhorn projection operator is not Lipschitz in the standard sense. This stands in contrast with Proposition 2, which illustrates that this projection is Lipschitz on the logarithmic scale.

This non-Lipschitz result holds even for the following simple rescaling of the $2 \times 2$ Birkhoff polyope:

$$\mathcal{M} := \left\{ P \in \mathbb{R}^{2\times 2}_{\geqslant 0} \ : \ P1 = P^T 1 = \begin{bmatrix} \frac{1}{2} \\ \frac{1}{2} \end{bmatrix} \right\}.$$

**Proposition 4.** *The Sinkhorn projection operator onto $\mathcal{M}$ is not Lipschitz for any norm on $\mathbb{R}^{2\times 2}$.*

*Proof.* By the equivalence of finite-dimensional norms, it suffices to prove this for $\|\cdot\|_1$, for which we will show

$$\sup_{K, K' \in \Delta_{2,2} \cap \mathbb{R}^{2\times 2}_{>0}} \frac{\left\| \Pi^{\mathcal{S}}_{\mathcal{M}}(K) - \Pi^{\mathcal{S}}_{\mathcal{M}}(K') \right\|_1}{\|K - K'\|_1} = \infty. \tag{9}$$

The restriction to strictly positive matrices $\mathbb{R}^{2\times 2}_{>0}$ (rather than non-negative matrices) is to ensure that there is no issue of existence of Sinkhorn projections (see, e.g., Linial et al., 1998, Section 2).

For $\varepsilon, \delta \in (0, 1)$, define the matrix

$$K_{\varepsilon, \delta} := \begin{bmatrix} 1 - \varepsilon & \varepsilon \\ 1 - \delta & \delta \end{bmatrix}$$

and let $P_{\varepsilon, \delta} := \Pi^{\mathcal{S}}_{\mathcal{M}}(K_{\varepsilon, \delta})$ denote the Sinkhorn projection of $K_{\varepsilon, \delta}$ onto $\mathcal{M}$. The polytope $\mathcal{M}$ is parameterizable by a single scalar as follows:

$$\mathcal{M} = \{ M_a \ : \ a \in [0, 1/2] \}, \qquad M_a := \begin{bmatrix} a & \frac{1}{2} - a \\ \frac{1}{2} - a & a \end{bmatrix}.$$

By definition, $P_{\varepsilon, \delta}$ is the unique matrix in $\mathcal{M}$ of the form $D_1 K_{\varepsilon, \delta} D_2$ for positive diagonal matrices $D_1$ and $D_2$. Taking

$$D_1 = \begin{bmatrix} \sqrt{\frac{\delta}{\varepsilon}} & 0 \\ 0 & \sqrt{\frac{1-\varepsilon}{1-\delta}} \end{bmatrix}, \qquad D_2 = \frac{1}{\beta_{\varepsilon,\delta}} \begin{bmatrix} \sqrt{\frac{\varepsilon}{1-\varepsilon}} & 0 \\ 0 & \sqrt{\frac{1-\delta}{\delta}} \end{bmatrix}$$

for $\beta_{\varepsilon,\delta} = 2(\sqrt{\delta(1-\varepsilon)} + \sqrt{\varepsilon(1-\delta)})$, we verify $D_1 K_{\varepsilon,\delta} D_2 = M_{a_{\varepsilon,\delta}}$, where $a_{\varepsilon,\delta} := \frac{\sqrt{\delta(1-\varepsilon)}}{\beta_{\varepsilon,\delta}}$. Therefore $P_{\varepsilon,\delta} = M_{a_{\varepsilon,\delta}}$ for $\varepsilon, \delta \in (0, 1)$.

Now parameterize $\varepsilon := c\delta$ for some fixed constant $c \in (0, \infty)$ and consider taking $\delta \to 0^+$. Then $a_{c\delta, \delta} = \frac{\sqrt{1-c\delta}}{2\left[\sqrt{c(1-\delta)} + \sqrt{1-c\delta}\right]}$, which for fixed $c$ becomes arbitrarily close to

$$g(c) := \frac{1}{2(\sqrt{c} + 1)}$$

as $\delta$ approaches 0. Thus $\|P_{c\delta,\delta} - M_{g(c)}\|_1 = o_\delta(1)$ and similarly $\|P_{\delta/c,\delta} - M_{g(1/c)}\|_1 = o_\delta(1)$. We therefore conclude that for any constant $c \in (0, \infty) \setminus \{1\}$, although

$$\|K_{c\delta,\delta} - K_{\delta/c,\delta}\|_1 = 2\delta \, |c - 1/c| = o_\delta(1),$$

vanishes as $\delta \to 0^+$, the quantity

$$\begin{aligned}
\left\|\Pi_{\mathcal{M}}^{\mathcal{S}}(K_{c\delta,\delta}) - \Pi_{\mathcal{M}}^{\mathcal{S}}(K_{\delta/c,\delta})\right\|_1 &= \left\|P_{c\delta,c} - P_{\delta/c,c}\right\|_1 \\
&= \left\|M_{g(c)} - M_{g(1/c)}\right\|_1 + o_\delta(1) \\
&= 4 \, |g(c) - g(1/c)| + o_\delta(1)
\end{aligned}$$

does not vanish. Therefore combining the above two displays and taking, e.g., $c = 2$ proves (9). $\qquad\square$

## Footnotes

[1]We use quotations since it is not technically a distance; see (Cuturi, 2013, Section 3.2) for details. The quotes are dropped henceforth.

[2]We are indebted to Piotr Indyk for inspiring this remark.

[3]Pseudocode for the variant we employ can be found in Appendix A.1.

[4]`http://graphics.stanford.edu/data/3Dscanrep/`

[5]`http://www.kernel-operations.io/geomloss/`