[Reviews · NeurIPS 2019]

Reviewer 1



The problem of estimating Sinkhorn distance has been well studied. Given two probability distributions p and q, both on the same metric space, the Sinkhorn distance is a regularized notion of how measuring how different the two probability distributions are. In general, calculating the Sinkhorn distance needs O(n^2) time, where n = support of p or q. As the authors discuss, there are efficient algorithms that utilize the structure of the metric space (e.g. it being low-dimensional) and try to make this more efficient. In this paper, the novel contribution is the following: the authors utilize the connection between Sinkhorn distance and Sinkhorn scaling to give an efficient algorithm. They then use Nystrom method to do Sinkhorn scaling. Again, while Nystrom methods have been used in this setting before, this presents a neat bound in terms of what they call "effective dimension". They are able to prove that their algorithm leads to a guaranteed bound on the Sinkhorn distance. They also perform experiments to demonstrate the efficiency. The experiments seems quite exhaustive in terms of the baseline algorithms compared against. My overall impression is the following: while a number of blocks that are used in the algorithm are existing in literature, putting them together and proving the theoretical bounds requires substantial insight. The authors also extend their bounds to special cases when the data lie in low-dimensional manifolds. The paper seems well written too. I was not entirely sure why we would expect R to appear in the run-time? Any intuition , e.g. whether it is inevitable or is a effect of this analysis, would be useful.

Reviewer 2



Comments: - interesting paper - please check the paper w.r.t. the usage of similarities, dissimilarities, distances and make it consistent - some assumptions like psd should be clearly mentioned - 'method constructs a low-rank approximation to a Gaussian kernel' - well technically it can be used for any psd kernel - and even more generic (non-psd) as shown in A. Gisbrecht et al. Metric and non-metric proximity transformations at linear costs. Neurocomputing 167: 643-657 (2015) - Adaptive Nystroem is fine but can lead to suboptimal results for very small set of landmarks hence the number of landmarks should be reasonable large -> which at the end has some negative impact on the memory / runtime consumption - Could you please also comment on the out of sample extension - e.g. if I use the calculated/approximated proximities and generate a model how should I effectively and valid apply the model on new data - would it be 'safe' to calculate the proximities exact? - it would be nice to have a larger evaluation on the practical impact (not only runtime) of the approximation on its usage in some algorithm (so far only Fig 2) --> this is somewhat covered in the supplement but its a bit sad not to have a clear summary in the main paper

Reviewer 3



This paper proposes simple method to approximate Sinkhorn distance quickly by combining two previous techniques and presents a new analysis. The experimental results also show that the algorithm can obtain a low excess error efficiently. The results seem convincing. The presentation could be improved. The background is not enough and some notations are used without definition.

[Author Response · NeurIPS 2019]

We kindly thank the reviewers for their positive comments and insightful suggestions. Detailed responses below.

Reviewer 1:

- *"I was not entirely sure why we would expect R to appear in the runtime? Any intuition, e.g. whether it is inevitable or is an effect of this analysis, would be useful."* This is a good question for which there is an intuitive answer: the dependence on the radius $R$ is inevitable for any algorithm which approximates OT to $\varepsilon$ additive error, essentially due to a scale-invariance argument. Specifically, since the transportation cost $\sum_{ij} P_{ij} \|x_i - x_j\|_2^2$ is part of the definition of the Sinkorn distance (see eq (1) in the paper), if all pairwise distances $\|x_i - x_j\|_2^2$ increase by factor of say 10, then so does the transportation cost. In other words, approximating Sinkhorn divergence to $\varepsilon$ additive error becomes harder as the radius $R$ increases. We will add a comment to the camera-ready version to clarify this.

Reviewer 2:

- We will be sure to revise the camera-ready version to be consistent with those terminologies.
- We will add a comment about the applicability of Nyström to other, more general kernels. However, the approximation guarantees in our analysis (in Section 3) are specifically tailored to the Gaussian kernel.
- *"Adaptive Nyström is fine but . . . . memory/runtime consumption.* Yes, definitely agreed. This is precisely the purpose of the guarantees we proved in Section 3, which give provable guarantees on the number of landmarks required to obtain a given approximation error of the kernel matrix. These guarantees are then used to prove Theorem 1 (our main result) which gives provable bounds on the memory/runtime consumption of our proposed algorithm Nys-Sink.
- *"Could you please also comment on the out of sample extension... would it be 'safe' to calculate the proximities exact?"* Yes, out-of-sample extension is totally doable with Nyström.
- *"It would be nice to have a larger evaluation on the practical impact (not only runtime) of the approximation on its usage in some algorithm (so far only Fig 2) –> this is somewhat covered in the supplement but its a bit sad not to have a clear summary in the main paper."* We will move some of the relevant material from the supplement, and make sure to clarify in the main text.

Reviewer 3:

- We will be sure to revise the camera-ready version to define those notations, and to make clearer the background on the Sinkhorn distance.
- The reason we only consider squared Euclidean cost is that it is only in this case that the corresponding kernel matrix $K$ is a Gaussian kernel matrix, which our algorithms and analysis heavily rely upon. Nevertheless, we emphasize that in many applications, especially in the field of computer graphics, image processing and simulation of physical systems, the 2-Wasserstein distance is a very common choice. It is also among the most well-studied choices in theory. Our focus on this distance is therefore of both theoretical and practical significance.

[Meta-Review · NeurIPS 2019]

The paper presents a connection between Sinkhorn distance and Sinkhorn scaling to give an efficient algorithm, and they then use Nystrom method to do Sinkhorn scaling. While Nystrom methods have been used in this setting before, the paper presents a nice bound in terms of and effective dimension.